

# 9-year spatial and temporal evolution of desert dust aerosols over South-East Asia as revealed by CALIOP

Emmanouil Proestakis[1,2], Vassilis Amiridis[1], Eleni Marinou[1,3], Aristeidis K. Georgoulias[4,5], Stavros Solomos[1], Stelios Kazadzis[6,7], Julien Chimot[8], Huizheng Che[9,10], Georgia Alexandri[4], Ioannis Binietoglou[11], Konstantinos A. Kourtidis[4], Gerrit de Leeuw[12,13], Ronald J. van der A[14]

[1]IAASARS, National Observatory of Athens, Athens, 15236, Greece
[2]Laboratory of Atmospheric Physics, Department of Physics, University of Patras, 26500, Greece
[3]Department of Physics, Aristotle University of Thessaloniki, Thessaloniki, 54124, Greece
[4]Laboratory of Atmospheric Pollution and Pollution Control Engineering of Atmospheric Pollutants, Department of Environmental Engineering, Democritus University of Thrace, Xanthi, Greece
[5]Energy, Environment and Water Research Center, Cyprus Institute, Nicosia, Cyprus
[6]Institute of Environmental Research and Sustainable Development, National Observatory of Athens, Lofos Koufou, 15236 Penteli, Athens, Greece
[7]Physikalisch - Meteorologisches Observatorium Davos, World Radiation Center (PMOD/WRC) Dorfstrasse 33, CH-7260, Davos Dorf, Switzerland
[8]Department of Geoscience and Remote Sensing (GRS), Civil Engineering and Geosciences, TU Delft, the Netherlands
[9]Key Laboratory of Atmospheric Chemistry (LAC), Chinese Academy of Meteorological Sciences (CAMS), CMA, Beijing, 100081, China
[10]Jiangsu Collaborative Innovation Center of Climate Change, Nanjing, 210093, China
[11]National Institute of R&D for Optoelectronics, Magurele, Romania
[12]Finnish Meteorological Institute (FMI), Helsinki, Finland
[13]Department of Physics, University of Helsinki, Helsinki, Finland
[14]Royal Netherlands Meteorological Institute, De Bilt, Netherlands

*Correspondence to:* Emmanouil Proestakis (proestakis@noa.gr)

**Abstract.** We present a 3-D climatology of the desert dust distribution over South-East Asia derived using CALIPSO (Cloud-Aerosol Lidar and Infrared Pathfinder Satellite Observation) data. To distinguish desert dust from total aerosol load we apply a methodology developed in the framework of EARLINET (European Aerosol Research Lidar Network), the particle linear depolarization ratio and updated lidar ratio values suitable for Asian dust, on multiyear CALIPSO observations (01/2007-12/2015). The resulting dust product provides information on the horizontal and vertical distribution of dust aerosols over SE (South-East) Asia along with the seasonal transition of dust transport pathways. Persistent high D_AOD (Dust Aerosol Optical Depth) values, of the order of 0.6, are present over the arid and semi-arid desert regions. Dust aerosol transport (range, height and intensity) is subject to high seasonality, with highest values observed during spring for northern China (Taklimakan/Gobi deserts) and during summer over the Indian subcontinent (Thar Desert). Additionally we decompose the CALIPSO AOD (Aerosol Optical Depth) into dust and non-dust aerosol components to reveal the non-dust AOD over the highly industrialized and densely populated regions of SE Asia, where the non-dust aerosols yield AOD values of the order of 0.5. Furthermore, the CALIPSO-based short-term AOD and D_AOD time series and trends between 01/2007 and 12/2015 are calculated over SE Asia and over selected sub-regions. Positive trends are observed over northwest and east China and the Indian subcontinent, whereas over southeast China are mostly negative. The calculated AOD trends agree well with the trends derived from Aqua/MODIS (Moderate Resolution Imaging Spectroradiometer), although significant differences are observed over specific regions.

## 1 Introduction

Airborne mineral dust is a major category of particles in the Earth's atmosphere that influences climate from local and regional to global scale (Huang et al., 2006). Dust aerosols have a significant role on climate through the direct radiative effect of absorption and scattering of solar and thermal terrestrial radiation (Ramanathan et al., 2001; Tegen et al., 1996). Moreover



dust aerosols, depending on the atmospheric conditions and on the dust composition, act either as effective CCN (Cloud Condensation Nuclei) (Hatch et al., 2008) or as IN (Ice Nuclei) (DeMott et al., 2009; Chou et al., 2011), modifying cloud albedo, coverage and precipitation (Rosenfeld et al., 2008). Hence, the indirect effect of dust on the Earth's climate lies in the modulation of the solar radiation forcing by altering the cloud microphysical and macrophysical properties (Twomey, 1977;

Albrecht, 1989; Haywood and Boucher, 2000). Besides the direct and the indirect effects and the effect on meteorological processes, dust transported over large distances has significant impact on human health and life expectancy due to degradation of air quality (Griffin, 2007; Goudie, 2014). In addition to human quality of life, the aeolian transport of dust is crucial for the sustainability of marine and terrestrial ecosystems through the deposition of mineral inputs and nutrients (Martin et al., 1994; Okin et al., 2004; Jickells et al., 2005).

Over Asia, airborne mineral dust is considered a significant atmospheric aerosol contributor. Major dust Asian sources include the deserts of the Arabian Peninsula in southwest Asia and the Middle East, Thar Desert (Pakistan/India), the sandy Taklimakan region across northwest China and the vast arid and semiarid Gobi desert in north China and southern Mongolia (Figure 1). The dust aerosol load generated in the Gobi and Taklimakan deserts is estimated to 800 Tg yr$^{-1}$ (Zhang et al., 1997). Airborne dust originating from Asian deserts is frequently transported eastward across China (Zhang et al., 2003), over the North Pacific

Ocean (Shaw, 1980; Duce et al., 1980) to the Western coast of North America (Uno et al., 2001; Huang et al., 2008) and in extreme cases to even longer distances, completing full global circuits (Clarke et al., 2001; Uno et al., 2009).

In order to examine the composition, properties and radiative effect of Asian dust, several field campaigns have been conducted. Regional aircraft and ground-based campaigns, such as the Indian Ocean Experiment (INDOEX) (Rasch et al., 2001), the Asian Aerosol Characterization Experiment (ACE-Asia) (Huebert et al., 2003), the Intercontinental Chemical

Transport Experiment (INTEX-B) (McKendry et al., 2008), the Aeolian Dust Experiment on Climate Impact (ADEC) (Mikami et al., 2006), the NASA Transport and Chemical Evolution over the Pacific (TRACE-P) (Jacob et al., 2003) and the Pacific Dust Experiment (PACDEX) (Stith et al., 2009) have contributed considerably to our knowledge and understanding of Asian dust. The conducted field campaigns were crucial for the investigation and understanding of Asian dust properties. Moreover, extensive measurements on the spatial and temporal variability and the evolution of dust aerosols are required in order to assess

their contribution to the climate. To this end, several passive remote sensing satellite instruments, such as the Advanced Very High Resolution Radiometer (AVHRR) (Husar et al., 1997), the MODerate resolution Imaging Spectroradiometer (MODIS) (Remer et al., 2005) on board both Terra and Aqua platforms, the Total Ozone Mapping Spectrometer (TOMS) on board Nimbus 7 (Prospero et al., 2002) and the Ozone Mapping Instrument (OMI) onboard Aura have been used (Chimot et al., 2017). Although passive satellite sensors provide information on the column properties of aerosols with adequate spatial and

temporal resolution, they are bound to certain limitations, the major limitation being the lack of information on the three-dimensional distribution (vertical profile) of aerosols in the atmosphere, an important information for the assessment of the aerosols radiative forcing on climate as well as their contribution as IN and CCN (IPCC 2013).

The vertical structure of aerosols and clouds is provided through ground-based and satellite-based LIDAR (LIght Detection And Ranging) systems (Liu et al., 2002). Regarding satellite-based lidar observations, CALIOP (Cloud Aerosol Lidar with

Orthogonal Polarization) provides vertically resolved information on both aerosols and clouds on a global scale since June 2006. CALIOP consists the main instrument onboard the NASA/CNES CALIPSO (Cloud-Aerosol Lidar and Infrared Pathfinder Satellite Observations) satellite (Winker et al., 2007). CALIOP measures total attenuated backscatters signals at 532nm and 1064nm and the linear depolarization ratio at 532nm. The depolarization ratio is a crucial parameter in the dust aerosol classification (Ansmann et al., 2003, Liu et al., 2008c), since dust particles are strongly depolarizing, as opposed to

other aerosol types. Thus CALIOP is an ideal instrument to detect dust aerosol and study its 3D spatial distribution and temporal evolution (Yang et al., 2012; Amiridis et al., 2013; Marinou et al. 2017).

Over the past decade CALIOP retrievals have been frequently utilized in dust aerosol studies focusing on SE Asia. For instance, Huang et al. (2007) examined summertime dust plumes appearing over the Tibetan Plateau and found that they originate from





the nearby Taklimakan Desert. Liu et al. (2008b) examined the spatial distribution of dust over the Tibetan Plateau and the surrounding areas on a seasonal basis. Huang et al. (2008), using CALIOP, a micropulse lidar and surface meteorological data from the Gansu Meteorological Bureau studied the long-range transport of dust from the Taklimakan and Gobi deserts over east China and the Pacific Ocean during the PACDEX (PACific Dust EXperiment) campaign (March to May 2007). They also

showed that the dust storms over the Gobi region are more intense but less frequent than the ones over the Taklimakan region. The passage of dust from Gobi to Japan (covering 1000-1500 km/day) and consequently over the Pacific Ocean was also reported by Uno et al. (2008) using extinction coefficients from CALIOP, a dust transport model and forward trajectory analysis. Huang et al. (2010) used CALIOP dust data along with other A-Train satellite observations to study the climatic effect of dust on the semi-arid areas of north-west China. Ge et al. (2014) combined dust CALIOP data with observations from

the Multiangle Imaging Spectroradiometer (MISR) showing that the Taklimakan dust can be lofted vertically up to 10 km height above sea level as a result of the local topography and synoptic conditions. He and Yi (2015) utilizing data from CALIOP and ground based lidar observations over China examined 13 dust events within the period 2010-2012, while Xu et al. (2016) studied the horizontal, vertical and temporal variability of dust aerosols over China based on CALIOP. Satellite-based observations from CALIPSO have also been utilized to study the effect of dust transport from Thar Desert to the Indian

subcontinent (Gautam 2009; Das et al., 2013; Kumar et al., 2014). More recently, Tan et al. (2016) and (2017) combined CALIPSO, MODIS-Terra, OMI data and ground-based dust records and studied the transport processes of five dust storms in spring 2010 from the Taklimakan and Gobi deserts to the Pacific and their impacts on the ocean.

The aforementioned studies used the standard CALIPSO product and aerosol subtype classification (Omar et al., 2009). In this way either considered as dust explicitly the classified as dust aerosol subtype, hence not taking into consideration the dust

component of the polluted dust aerosol subtype, or utilized both the dust and polluted dust aerosol subtypes, hence taking into consideration the non-dust component of polluted dust. In the present study we use a separation methodology developed in the framework of EARLINET (European Aerosol Research Lidar Network), the particle linear depolarization ratio and updated lidar ratio values suitable for Asian dust, in order to distinguish the pure dust component from the dust and polluted dust aerosol subtypes. In this paper we use the pure dust product in order to provide the three-dimensional seasonal distribution and

the short-term temporal evolution of dust over SE Asia, based on nine years of CALIOP observations (01/2007-12/2015). The domain of the study is confined between 65°-155° E and 5°-55° N (Fig 1). The pure dust product can be used in the evaluation of models related to dust transport and to radiative transfer models, in studies of dust-related physical processes (dust transport dynamics, CCN, IN), to investigate the effect of dust aerosols on ecosystems (dust deposition into the oceans) and to determine the dust aerosol load over highly industrialized and densely populated regions.

The paper is organized as follows: Sect. 2 provides a description of CALIOP, of the data used and the methodology followed in the study. Sect. 3 provides the main results. In Sect. 3.1 results on the horizontal distribution of aerosols over SE Asia (AOD, D_AOD, non-dust AOD) and of the observed dust Center of Mass and dust Top Height over SE Asia are presented and discussed. The vertical distribution of dust aerosols is presented and discussed in Sect. 3.2-3.3 through the dust climatological and conditional extinction coefficient profiles, while the short-term temporal evolutions of AOD and D_AOD during the study

period are examined in Sect. 3.4. Finally, Sect. 4 provides a summary of the study along with the main concluding remarks.

## 2    Data and methodology

CALIPSO is a sun-synchronous polar orbit satellite with an equatorial crossing time around 13:30 local time and approximately sixteen days repetition orbit. CALIPSO, the collaborative NASA and CNES project, joined the A-Train formation of satellites on April 2006 (Winker et al. 2007). CALIOP, the primary instrument onboard CALIPSO, consists of an elastic backscatter

and polarization Nd:YAG laser (Hunt et al., 2009). CALIOP transmits linear polarized light, while a telescope (1m diameter) collects the backscatter component by the atmosphere. Utilizing the total backscatter signals and the polarization of the



backscattered light CALIOP provides almost continuously height-resolved information on the vertical structure of aerosols and clouds (Winker et al., 2009), from the ground up to 30 km height.

Three levels of CALIPSO products are provided from NASA and CNES. The Level-1 (L1) product consists of the raw range corrected signals in the highest spatio-temporal resolution. The Level-2 (L2) product consists the high-level quality products.

More specifically, CALIPSO L2 algorithm classifies the detected layers into characteristic classes (Vaughan et al., 2009). The atmospheric features types classified as aerosols are further distinguished into specific aerosol subtypes (Clean Marine, Dust, Clean Continental, Polluted continental, Polluted Dust and Smoke). The classification algorithm (Omar et al., 2009) utilizes the depolarization ratio and the magnitude of the attenuated backscatter signal, the height of the aerosol layers and the characteristics of the Earth's surface along the CALIPSO footprint (desert, ocean, snow/ice) in order to discriminate the

detected atmospheric features types into subtypes. In addition, CALIPSO L2 algorithm uses specific lidar ratios (LR) for each classified aerosol type in order to derive the profiles of extinction coefficient (Young and Vaughan, 2009). The final L2 product is characterized by 5 km horizontal resolution and vertical resolution of 60m in the altitude range -0.5-20.2 km and 120m in the altitude range 20.2-30.1 km above sea level. The L2 aerosol extinction product is used to provide the Level 3 (L3) product of CALIPSO, characterized by 2ox5o grid resolution (Winker et al., 2013).

In the framework of this study we use the CALIPSO L2 optimized profiles (based on the CALIPSO Version 3 dataset), developed as an intermediate product under the collaborative EARLINET-ESA LIVAS (LIdar climatology of Vertical Aerosol Structure) project (Amiridis et al., 2015). This product has a spatial resolution of 1ox1o and is described in detail and compared with AERONET in Amiridis el at. (2015). In brief, for this product, several quality control filters are applied in the CALIPSO L2 V3 dataset, following the filtering proposed for L3 product (Winker et al. 2013). Moreover, in order to ensure the high-

quality of the aerosol product, in addition to the filters described in Winker et al. (2013), L2 profiles with cloud observations are filtered from the dataset (Amiridis et al., 2013).

In addition to the CALIPSO L2 optimized profiles, the aerosol observations categorized through the CALIOP classification scheme as dust or polluted dust (Omar et al., 2009) are used in order to retrieve the pure dust aerosol component. To this end, the particle depolarization ratio of dust is used. During SAMUM 1 and 2 campaigns Saharan dust particle depolarization ratio

values varied between 0.27 and 0.35 at 532 nm (Ansmann et al., 2011). Typical dust particle depolarization ratio values measured with lidars in field campaigns around the globe are consistent with this values, showing little variation independently of the source region, (e.g., Sakai et al., 2000; Liu et al., 2008b; Freudenthaler et al., 2009; Groß et al., 2011; Burton et al., 2013; Groß et al., 2013; Groß et al., 2015; Illingworth et al., 2015). According to the methodology proposed by Tesche et al. (2009) the aerosol layers classified as dust or polluted dust also having a depolarization ratio lower than 0.31 are assumed to

be a mixture of pure dust and non-dust aerosol components. The particle depolarization ratio value of the pure dust component is then calculated by:

$$\delta_p = \frac{\beta_\perp}{\beta_t - \beta_\perp} \qquad [1]$$

where $\delta_p$ is the particle depolarization value of the pure dust component, $\beta_\perp$ is the perpendicular component of the backscatter value and $\beta_t$ is the total backscatter of the aerosol layer. The backscatter contribution of the pure dust component is calculated by:

$$\beta_1 = \beta_t \frac{(\delta_p - \delta_2)(1 + \delta_1)}{(\delta_1 - \delta_2)(1 + \delta_p)} \qquad [2]$$

where $\delta_1$ ($\delta_2$) is a theoretical depolarization value of the dust (non-dust) component. For the non-dust aerosols, we assume particle depolarization ratio values of 0.03, considering minor contributions to depolarization by dried marine particles and by anthropogenic particles. Using this methodology, the CALIPSO pure dust backscatter coefficient profile at 532nm is calculated.



Regarding the aerosol transport, the intercontinental transport depends on the atmospheric circulation, which in the case of Asia is heavily affected by the Himalaya orographic barrier. The area SW of Himalayas (Arabic Peninsula, India, Indian Ocean) is mostly affected by long range transport of dust generated from the Arabian Desert and from the arid areas of Somalia and Ethiopia (Prospero et al., 2002). Local dust sources in arid areas of Iran, Iraq and Afghanistan additionally contribute to

the regional dust load. On the contrary, the areas located to the east of the Himalaya barrier (Mongolia, China, SE Asia Peninsula, Pacific Ocean) are mostly affected by dust originating from the Gobi and Taklimakan deserts (Prospero et al., 2002). In order to retrieve the pure dust extinction coefficient profile at 532nm the pure dust backscatter coefficient profile has to be multiplied with the appropriate LR for Asian dust. The LRs observed globally are summarized in the works of Müller et al. (2007) and Baars et al. (2016). In general, different desserts produce dust with different properties, thus with different LRs.

Typical values of LRs of desert dust vary between 35 and 55 sr. The LR of desert dust originating from deserts of the Arabian Peninsula, the Taklimakan region and the vast semiarid Gobi desert has been investigated with ground based lidars (Sakai et al., 2002; Murayama et al., 2004; Ansmann et al., 2005; Tesche et al., 2007; Xie et al., 2008; Haenel et al., 2012; Komppula et al. 2012; Mamouri et al., 2013), airborne instrumentation (Anderson et al., 2003) and during intensive campaigns (Liu et al., 2002, Murayama et al., 2003). Based on the atmospheric circulation over Asia, the dust aerosol transport and the observed

LRs, the domain of this study can be divided in two subdomains and two different LRs for pure dust can be assigned in these regions. South-west of the Himalayas (Arabic Peninsula, India, Indian Ocean) a LR of 40sr is assigned to pure dust, while east of the Himalayas the value of 47sr is used. The assigned LR values are used for the retrieval of the pure dust aerosol extinction coefficient profiles at 532nm through the backscatter coefficient profile at 532nm (Tesche et al., 2009).

In this study, based on the aerosol extinction coefficient profiles at 532nm and on the pure dust extinction profiles at 532nm

(from here on referred as dust extinction), the following products are discussed:

The seasonal CALIPSO L3 optimized Aerosol Optical Depth (AOD), dust Optical Depth (D_AOD) and non-dust Aerosol Optical Depth aggregated in 1ox1o spatial resolution grids.

The seasonal CALIPSO L3 dust profile Top Height (TH) and dust Center of Mass (CoM) aggregated in grids of 1°x1° spatial resolution. The dust TH (km) is defined as the height where 98% of the D_AOD lies below while the dust CoM height is

defined according to Mona et al. (2006), as the backscatter weighted altitude where 50% of the D_AOD lies below. The CoM is given by the equation:

$$\text{CoM} = \frac{\int_{z_b}^{z_t} z \cdot \beta(z)\,dz}{\int_{z_b}^{z_t} \beta(z)\,dz} \quad [\text{km}] \qquad\qquad [3]$$

where: z is the height in the atmosphere, $\beta(z)$ is the extinction coefficient of the dust layer at heights z, $z_b$ and $z_t$ the base and top heights of the profile respectively.

The seasonal zonal distribution of the climatological and conditional dust extinction coefficient (Mm$^{-1}$). The climatological

dust product is a measure of the average dust load over a geographical domain and is computed acknowledging only the contribution of the dust component in the atmosphere. This is accomplished by setting the dust extinction coefficient value of 0 km$^{-1}$, for observations with non-dust aerosols. The conditional dust product is a measure of the average intensity of dust load over a geographical domain and is based explicitly on the dust profiles, hence ignoring completely non-dust aerosols. In the recent study of Marinou et al. (2017), the climatological and conditional dust products have been used to study the dust

distribution above Europe and North Africa. In this study, the climatological and conditional products are similar to the study of Marinou et al., (2017), with the only difference in the selection of the domain depended LR values (55sr for Saharan desert). Evaluation of this product against collocated AERONET observations show absolute biases between CALIPSO and AERONET AODs of -0.03 (Amiridis et al., 2013).

Short-term CALIOP time series and trends of AOD and D_AOD for the study domain, based on nine years of CALIPSO

overpasses (01/2007-12/2015). In addition to the CALIPSO/CALIOP time series and trends, Aqua/MODIS trends for the same period (01/2007-12/2015) are included.



The (Moderate resolution Imaging Spectroradiometer) MODIS onboard the EOS Aqua satellite was launched on 4 May 2002. The sensor has a daytime equator crossing at 13:30 LT (noon). Due to its wide swath (2330 km) it is capable of providing almost global coverage on a daily basis. MODIS measures backscattered radiation in 36 spectral bands, from 0.415 to 14.235 μm, with a spatial resolution of 250, 500 and 1000 m depending on the band. In this work, monthly AOD550 data from the

Level-3 MODIS/Aqua Collection 6 1°x1° gridded dataset (MYD08_M3) are used. The MODIS data were acquired from NASA's Level 1 and Atmosphere Archive and Distribution System (LAADS) (http://ladsweb.nascom.nasa.gov) covering the period from 01/2007 to 12/2015. AOD550 is retrieved using two algorithms, the "Dark Target" (DT) and "Deep Blue" (DB). There are two separate DT algorithms, one used for land surfaces and one for water surfaces (Kaufman et al., 1997; Tanré et al., 1997; Remer et al., 2005; Levy et al., 2010, 2013). The DT expected error for Collection 6 is $\pm(0.05+0.15\tau_A)$ over land and

$+(0.04+0.1\tau_A)$, $-(0.02+0.1\tau_A)$ over ocean relative to the AERONET aerosol optical thickness ($\tau_A$) (Levy et al., 2013). While DT is used over vegetated surfaces and surfaces covered by dark-soil, the DB algorithm is capable of retrieving AOD550 over bright surfaces such as deserts, arid and semiarid areas (Hsu et al., 2004, 2013). The new DB algorithm which is used for the production of Collection 6 is applicable over all land surfaces (Sayer et al., 2013, 2014; Hsu et al., 2013). For Collection 6 the DB expected error is $\sim\pm(0.03+0.2\tau_M)$ relative to the MODIS aerosol optical thickness ($\tau_M$) (Hsu et al., 2013; Sayer et al., 2015).

In this work, $AOD_{550}$ data from the merged (DT and DB) (Levy et al., 2013) datasets are used

Regarding the uncertainties of the products, CALIOP L2 V3 is characterized by daytime minimum detectable backscatter of $0.0017\pm0.0003$ km$^{-1}$sr$^{-1}$, nighttime minimum detectable backscatter of $0.0008\pm0.0001$ km$^{-1}$sr$^{-1}$ and minimum detectable AOD of 0.005 (Winker et al., 2009). The reported underestimation in the CALIPSO AOD (Kittaka et al., 2011; Rogers et al., 2014; Papagiannopoulos et al., 2016; Tian et al., 2017) is additionally related to the limitation of CALIOP to collect backscatter

signals lower than the minimum detectable backscatter from aerosol layers in the free troposphere. The estimation of the uncertainties of CALIPSO L2 V3 product is based on the assumptions that they are random and uncorrelated (Young, 2010). Under these assumptions the backscatter, depolarization and AOD are characterized by uncertainties of 30-100%, 30-160% and >100% respectively. In addition to the inherited uncertainties of the CALIOP L2 V3 standard product to the AOD and D_AOD optimized products used in the study, uncertainties are introduced due to the selection of LR values suitable for Asian

dust. A LR of 47±4 sr is used for dust aerosols emitted from the Taklimakan and Gobi deserts, based on the literature (Liu et al., 2002; Sakai et al., 2002; Anderson et al., 2003; Murayama et al., 2003; Murayama et al., 2004; Ansmann et al., 2005; Tesche et al., 2007; Xie et al., 2008; Hänel et al., 2012; Komppula et al. 2012; Mamouri et al., 2013). The LR introduces uncertainty in the D_AOD product approximately of 20%. Additional, uncertainty which propagates into the D_AOD product is introduced due to the depolarization ratio of the non-dust aerosols, coupled into the polluted-dust aerosol subtype. As already

discussed, for the non-dust component a mean value for the different aerosol subtypes depolarization ratio of 0.03 is used. Extended analysis on the way that uncertainties propagate into the products is presented in Marinou et al. (2017).

### 3    Results and discussion

#### 3.1  Horizontal distribution of aerosols and dust

In this section we present and discuss the CALIPSO L3 optimized AOD and the D_AOD products. Since the mechanisms of

dust generation and transport and the removal processes of aerosols from the atmosphere vary with season the seasonal approach is selected. The seasons are defined as follows: December-January-February (DJF), March-April-May (MAM), June-July-August (JJA) and September-October-November (SON). Figure 2 shows the spatial distribution of the seasonal-mean AOD (Fig. 2a, e, i, m), D_AOD (Fig. 2b, f, j, n), ΔOD computed as the difference between the total AOD and D_AOD (Fig. 2c, g, k, o) and the corresponding percentage of D_AOD to the total AOD (Fig. 2d, h, l, p) in 1ox1o grid resolution and based

on 9 years of CALIPSO observations (01/2007-12/2015).





Regarding the horizontal distribution of AOD and non-dust AOD, similar geographical patterns are evident between all four seasons, although the observed features vary in magnitude. High values of non-dust AOD are consistently observed over the heavily industrialized and densely populated regions of India, Bangladesh and China. Over the Indo-Gangetic Plain and the entire region extending between New Delhi and Calcutta, the observed non-dust AOD values are persistently higher than 0.5

during DJF (Fig. 2a-d), MAM (Fig. 2e-h) and SON (Fig. 2m-p) while during JJA (Fig. 2i-l) the AOD is suppressed (< 0.3). The relatively lower AOD values observed during JJA over the Bay of Bengal (0.3) is related to the regional meteorology (monsoons). The high frequency of cloudiness results in weighted mean AOD values since extreme aerosol events are less frequently captured (Winker et al. 2013). Furthermore the wet deposition rate of aerosol increases during the summer monsoon period of the year (Lau e t al., 1988). The countries of Indochina are characterized by inhomogeneities in the observed aerosol

load, with larger non-Dust AOD values observed during MAM (> 0.5) and DJF (> 0.3) and lower values during JJA (< 0.15) and SON (< 0.1). The lower values during these months are attributed to the monsoon season in the area, which runs roughly between June and October. Over the Maritime Southeastern Asia, the AODs are relatively similar between the different seasons with mean AOD values of 0.2±0.1. Similar features have been shown by Campbell et al., (2013) who investigated the particle presence and the 2D variability of aerosol over the Indonesia region. Over China similar geographical patterns in the horizontal

distribution of aerosols are evident between all four seasons, with larger non-dust AOD values over the major sources of anthropogenic aerosols (Beijing, Shanghai, Guangzhou, Chongqing, Wuhan) such as urban clusters (Kourtidis et al., 2015) and D_AOD values over the deserts of Taklimakan and Gobi (Che et al. 2014, 2015).

Regarding the horizontal distribution of dust aerosols over SE Asia, the main difference is attributed to the high seasonality of dust aerosols generation and transport. Moreover, the activation mechanisms of the desert regions may vary as well (Prospero

et al., 2002). Asian dust emission sources in India (Thar Desert) and China (Taklimakan, Gobi) are clearly mapped through the systematic high D_AOD values throughout the year (Fig. 2b, f, j, n). The seasonality of the Great Arabian Desert, the Thar Desert and the arid regions of Ethiopia and Somalia is mainly related to the West Indian Monsoon activity (Vinoj et al., 2014) and is mostly evident during summer months (Fig. 2f). The local dust sources at the arid areas of Iran, Iraq and Afghanistan additionally contribute to the regional dust load. However the activation mechanism of these sources is mainly related to

convective episodes (Karami et al., 2017) and the contribution of these dust events to long range transport is limited. On the contrary to the desert regions of SW Asia, the maximum activity of Gobi and Taklimakan deserts is observed during March and May (Husar et al., 2001).

Regarding the transport of dust aerosols, the long-range transport is usually related to the activation of the major deserts (Liu et al., 2008a). Dust aerosols emitted from the Great Arabian Desert, Thar Desert and Somalia are transported eastwards over

India and the Indian Ocean reaching even the west coast of Indochina and Indonesia (Mao et al., 2011). The feature of dust transport over the Indian Peninsula and the Bay of Bengal is more prominent during MAM and JJA (Fig. 2f, j). The transported dust aerosols significantly contribute to the observed aerosol load over India, although the magnitude of the contribution varies per season. Over N. China a similar pattern of a persistent dust aerosol background is evident during all seasons, with a peak during MAM (Fig. 2f). The Asian dust generated from the Gobi and Taklimakan deserts is transported over China, Korea and

Japan and across the Pacific Ocean (Liu et al., 2008b). This dust belt is usually confined between 25ºN and 45ºN (Fig. 2d), extends frequently towards the western coast of N. America and is most prominent during MAM (Clarke et al., 2001; Uno et al. 2009).

Regions of low AOD and D_AOD values, regardless of the season of the year, are additionally evident. On climatological basis little evidence of dust transport over the Himalayas orographic barrier and low AOD over the Tibetan Plateau are shown.

This is in line with previous studies, reporting rare events of dust transport over Himalayas (Huang et al., 2007; Liu et al., 2008b; Yumimoto et al., 2009). The region to the North of Taklimakan, Gobi and Mongolia is also characterized by low values of AOD and D_AOD, except during MAM, indicating the prevailing eastward motion of both dust and anthropogenic aerosols.





Similarly, the maritime region of the Pacific Ocean south of 25ºN is also characterized by low AOD and D_AOD values, observation which is in line with previous studies (Huang et al., 2008; Kellogg and Griffin, 2006).

Figure 3 shows the seasonal geographical distribution of dust occurrences (Fig. 3a, d, g, j), dust CoM (Fig. 3b, e, h, k) and the corresponding Dust TH (Fig. 3c, f, i, l). The dust occurrences are calculated as the number of CALIPSO overpasses with Dust

observations, compared to the total number of CALIPSO overpasses (percentage). Both the dust CoM and TH are defined as the height in km above surface elevation (a.s.e.). The dust occurrences, dust CoM and dust TH are provided in a spatial resolution of 1ºx1º deg and are based on nine years of CALIOP observations, between 01/2007 and 12/2015.

The distributions of dust occurrences show that over the extensive desert areas of SE Asia (Tarim Basin, Thar Desert, southern Mongolia and Pakistan) the presence of dust is continuously high, over 80%, throughout the year. Between Taklimakan, Gobi

and Thar deserts similar seasonal features are observed. Based on Figure 3, the occurrence of dust over these desert regions reach a maximum during spring (Fig. 3d), while minimum dust activity is observed during winter (Fig. 3a). Lower frequencies of dust occurrence, which still exceed 70%, are also evident over east China and southeastern India. Conversely over Indochina and Indonesia the occurrence of dust is particularly low, particularly during summer (Fig. 3g) and autumn (Fig. 3j). The patterns of the dust frequency are in good agreement and consistent with the distribution of dust provided by Liu et al. (2008a), based

on one year of CALIPSO overpasses. Another noticeable feature of Figure 3 is the distinct pathways which are observed, a trans-Pacific belt between 25º and 45º N, and a second one over the Indian subcontinent towards the Bay of Bengal and the Arabian Sea. The observed values of dust occurrence over the major pathways decrease with increasing distance from the dust source regions. Furthermore, the distributions of dust occurrences show that the range of dust transport is subject to high seasonality. Over the dust belt of the Pacific Ocean (25º-45º N), values of dust occurrence vary between 30% during summer

(Fig. 3g) and 90% during spring (Fig. 3d). To the south of the dust belt and over the Pacific persistent low values of dust occurrence, which rarely exceed 30%, are observed almost all year long. The low dust occurrence over the Pacific Ocean south of 25º N concurs with studies based on CALIPSO regarding the long-range airborne transport of Asian dust (Huang et al., 2008; Liu et al., 2008a).

The distributions of dust CoM and TH show that during DJF (Fig. 3b,c) dust aerosols are in general suppressed below 3 km

height., with the CoM below 2 km. Maximum values of dust TH during DJF are observed across central and eastern China with a peak around 3 km height. During MAM (Fig. 3e,f) a large "dust belt" is observed, extending from the desert regions of Taklimakan and Gobi to the east across central China and over the Pacific Ocean. Dust is advected from the deserts, which are at elevation 1.5-2 km a.s.l. (Taklimakan) and 1-1.5 km a.s.l. (Gobi). When dust is transported eastward over the Pacific Ocean the above sea level distance of the layer remains constant, although due to the change of the surface elevation the absolute

distance above surface level seems to increase. Thus, the observed differences of CoM and TH between land and ocean and the high values observed over the Pacific Ocean, more pronounced during MAM, are an artifact of the change in the terrain elevation. In addition, the observed gradient in the horizontal distribution of D_AOD values between the sources and the Pacific Ocean (Fig. 2f), parallel to the ubiquitous dust layer and the high dust TH (Fig. 3f), are an indicator of the longer range of transport of lower concentration of dust particles. Over the Indian Peninsula and the Arabian Sea dust CoM (TH) tends to

be observed between 1-2 km (3-4 km) a.s.e., while both dust TH and dust CoM are decreasing toward southeast and over the Bay of Bengal. Himalaya Mountains are clearly observed as they act as a physical barrier to the transport of dust aerosols emitted from the Great Arabian and Thar deserts towards the Tibetan Plateau. Another noticeable feature of Figure 3 is that the dust aerosols observed over the Tibetan Plateau during MAM are transported from Taklimakan desert. During JJA, however, the pattern reverses (Fig. 3i). Dust CoM and TH values are larger over the Indian Peninsula and the active source of

Thar Desert than over the Taklimakan and Gobi deserts. The dust TH over Thar Desert extends to altitude as high as 5 km a.s.e. while over the Taklimakan and Gobi deserts the corresponding altitude varies around 3.5 km height. Furthermore during JJA dust aerosols emitted from the deserts to the SW of Himalayas, are transported into longer distances and injected into higher altitudes. The dust TH over the entire Indian Peninsula during summertime is in general around 3.5 km, gradually





decreasing eastwards towards the Bay of Bengal. To the North of Himalayas, over the Tibetan Plateau and central China large inhomogeneities in dust CoM and TH are observed. The "dust belt" during MAM is still evident, although the magnitude and extend of the dust transport is clearly decreased. The dust TH during JJA varies over China between 1.5 km to the coastal region and 4 km a.s.e. over the central China. The season with the minimum observed dust CoM and TH values throughout

the year is SON (Fig. 3k,l), with dust CoM and TH values between 1.5 and 3 km a.s.e.

A regional statistics description of the dust product is provided for six regions of interest over the domain of SE Asia: Indian Peninsula (5º-30ºN, 65º-95ºE), Tibetan Plateau (30º-36ºN, 80º-103ºE), Taklimakan / Gobi Deserts (36º-45ºN, 77º-115ºE), SE China, N Pacific Ocean (20º-45ºN, 125º-155ºE) and the Indochina / Indonesia region (5º-20ºN, 95º-155ºE). Figure 1 provides a map of the selected domains while the statistical description of the dataset per domain is provided in Table 1. More

specifically, Table 1 provides the mean D_AOD and standard deviation of D_AOD, the maximum observed D_AOD value and the 95[th] percentile, the mean dust CoM and the standard deviation of CoM in km (a.s.e.) and the dust mean TH and the standard deviation of dust TH in km (a.s.e.). Finally the number of profiles where dust or/and polluted dust aerosol subtypes to the total number of cloud-free profiles is included. The statistical representation of the dataset is provided per domain and per season, for the period 01/2007-12/2015.

### 3.2 Climatological dust extinction coefficient

In this section we present and discuss the vertical distributions of dust aerosols in the atmosphere over SE Asia, thus we present and discuss the vertical dimension of the 3-D dust distribution and transport. The derivation of a pure dust product from the CALIOP signal is of particular importance particularly for the densely populated areas of India and China where a significant percentage of the overall observed AOD is related to dust. The term climatological refers to the computation process, where

the mean pure dust extinction coefficient value is computed based on the cases where dust aerosols are detected, while the extinction coefficient of non-dust aerosol types is assigned to 0 $Mm^{-1}$. Therefore the zonal vertical distributions discussed in this section correspond to the horizontal distribution of the D_AOD presented in Sect.3.1.

The domain of interest, between 5° and 55° N, is divided into five 10° longitudinal bands. Based on 9 years of CALIPSO observations (01/2007-12/2015) Figure 4 shows the vertical distributions of the dust climatological extinction coefficient ($Mm^{-1}$),

defined in Section 2, for the four seasons. The surface elevation of the area is denoted with black colour in the plots (below the minimum elevation the contour plots have black colour). The continuous and dashed lines correspond to the average elevation of the surface level and to the average maximum elevation of the surface elevation respectively. A threshold of four dust cases is applied to the computation process of the pure dust climatological extinction coefficient, selected in order to avoid presenting extreme rare events in high altitudes at the same time with climatological values close to the surface level.

The North of the study domain, i.e. the region between 45º and 55º N (Fig. 4a, b, c, d), is characterized by relatively low values of dust extinction coefficient. Dust layers are relatively homogeneous and constrained below 4 km a.s.l. The dust climatological extinction coefficient values are in general below 25 $Mm^{-1}$. The highest values in this region, as high as 35 $Mm^{-1}$ are observed during DJF over the Plains of Manchuria (120º-135ºE) and extend as high as 5 km a.s.l. (Fig. 4a).

To the south of this region the Taklimakan and Gobi deserts (77º-115ºE) are the dominant land characteristics of the domain

between 35º and 45º N (Fig. 4e, f, g, h). Over this belt dust is ubiquitously present close to the surface throughout the year. Regarding the Taklimakan Desert, this region is a very prolific arid area encompassed by Tarim Basin. Due to the local topography of the Tarim Basin and the cyclonic systems generated over the Mongolia Plateau (Sun et al., 2001; Gong et al., 2006) Taklimakan dust activity is persistent active throughout the year (Liu et al., 2008b). Over this region the favourable topographic and meteorological conditions form an elevated layer of dust aerosols where climatological dust extinction

coefficient values greater than 100 $Mm^{-1}$ are regularly observed. The vast semiarid region to the East of the Tarim Basin, the Gobi Desert, is considered an additional source of Asian dust. Although the Tarim Basin is the primary source of Asian dust



to the North of the Tibetan Plateau, values of climatological dust extinction coefficient as high as 100 Mm⁻¹ close to the surface of Gobi Desert are present throughout the year. During the period between March and May the strong surface winds which develop over the Mongolian Plateau create favourable mechanisms of extreme dust events (Bory et al., 2003; Yu et al., 2008). More specifically, the maximum dust climatological extinction coefficient values over the entire SE Asia are observed over

the region of Taklimakan Desert during spring, reaching values as high as 200 Mm⁻¹ (Fig. 4f). Although the dust layer is mostly observed between 1.5 and 4 km a.s.l., during MAM lofted dust layers are detected as high as 9 km (a.s.l.). The observed features of dust transport are consistent with the values of elevation height reported in the literature (Huang et al., 2008; Eguchi et al., 2009). The elevated dust layers are captured by the strong westerly jet in the upper troposphere and accordingly transported eastwards across the mainland of China (Zhang et al., 2003) and the Pacific Ocean (Duce et al., 1980; Shaw, 1980).

This feature is evident throughout the year, although more pronounced during spring. The maximum height of dust transport also varies significantly with season. Moving from the Taklimakan Desert towards the coastline and over the Pacific Ocean, the highest altitude where dust layers are observed is decreasing from 8 km a.s.l. over the Taklimakan Desert to less than 2.5 km a.s.l. over the Pacific Ocean. The decrease of the altitude of transport of the dust layers is attributed to the gravitational settling of aerosols and to dry and wet deposition (Colarco et al., 2003). This characteristic is evident through the steep decrease

across the coastline and over the Yellow Sea and the Pacific Ocean, although during MAM a lofted layer of dust climatological extinction coefficient up to 25 Mm⁻¹ is observed up to 10 km a.s.l. (Fig. 4f). Close to the surface, over the densely populated and highly industrialized provinces of East China, a persistent dust layer with climatological extinction coefficient values as high as 100 Mm⁻¹ is observed throughout the year.

The region between 25° and 35° N, hence the area to the south of the dust belt which encompasses the deserts of Taklimakan

and Gobi, is the domain of Asia which is heavily affected by the Himalaya orographic barrier and the Tibetan Plateau (Fig. 4i, j, k, l). Dominant sources of dust aerosols of this area are the Thar Desert and the Arabian Peninsula to the west of Himalayas, while to the south lies the dense populated Indian subcontinent. Over the Thar Desert dust is ubiquitously present throughout the year, although the magnitude of dust activity is characterized by high seasonality. During the dry season, between March and May, an elevated layer of dust aerosols forms in the lower altitudes over Afghanistan, Pakistan and the western part of

India (Fig. 4j). Typical values of the dust climatological extinction coefficient are around 100 Mm⁻¹ and high concentrations of dust are observed close to the surface, although airborne dust is also frequently observed as high as 4 km altitude (a.s.l.). Significantly higher dust climatological extinction coefficient values, as high as 200 Mm⁻¹ are observed over Thar Desert during the summer season (Fig. 4k). Although the dust layer is primary observed between the surface and 2.5 km altitude, during JJA elevated layers of dust are detected over the sources at altitudes as high as 7 km. The elevated layer of dust is

accordingly transported eastwards, over the highly industrialized and densely populated Indo-Gangetic plains, where dust interacts with locally generated aerosol particles (Middleton, 1986). Due to the gravitational settling and to the dry and wet deposition (Colarco et al., 2003), significantly lower values are recorded southeastern of the dust sources, over the Indian subcontinent. The observed dust climatological extinction coefficient values range between 50 and 100 Mm⁻¹ over the Indo-Gangetic plains and the foothills of Himalayas. Dust climatological extinction coefficient values of 25 Mm⁻¹ indicate the

advection and presence of dust aerosols even as high as the Tibetan Plateau and the Himalayas. The observations regarding the vertical structure of dust over this domain support the Elevated Heat Pump hypothesis (Lau et al., 2006) of the accelerating Himalayas warming (Liu and Chen, 2000; Thompson et al., 2003) due to the presence of dust aerosols coupled with black carbon over the Tibetan Plateau. Significantly lower values, between 25 and 75 Mm⁻¹ are observed to the East of the Himalayas. The decrease in the dust climatological values over the Tibetan Plateau is less pronounced during MAM, when Taklimakan

and Gobi deserts to the North of this domain are characterized by maximum dust activity (Fig. 4j). Additionally, throughout the year, a steep decrease in the dust climatological extinction coefficient close to the coastline and the Pacific Ocean is evident. Over SE China the values close to the densely populated surface are persistently higher than 45 Mm⁻¹, while over the Pacific Ocean the dust climatological extinction coefficient values are decreased to less than 10 Mm⁻¹.





The domain between 15° and 25° N encompasses the largest part of India and of the countries of Indochina and Maritime Southeastern Asia (Fig. 4m, n, o, p). This domain is characterized by large inhomogeneities. High values of dust climatological extinction coefficient are observed over India and the Arabian Sea, as high as 100 Mm$^{-1}$, especially during MAM and JJA and lower values (Fig. 4n, o), below 50 Mm$^{-1}$, during SON and DJF (Fig. 4p, m). Over the Bay of Bengal the dust climatological

extinction coefficient values are drastically decreased compared to the mainland India and values around 25 Mm$^{-1}$ are frequently encountered. The steep decrease over the Bay of Bengal during MAM and JJA is most probably caused by wet deposition of dust aerosol particles due to the heavy monsoon rainfall (Lau et al., 2006).

Similar patterns are observed in the domain between 5° and 15° N, although the features vary in magnitude (Fig. 4q, r, s, t). Over the Arabian Sea, to the South of 15° N, values of dust climatological extinction coefficient around between 75 and 100

Mm$^{-1}$ are observed during MAM and JJA (Fig. 4r, s). Over South India during JJA elevated dust is present at altitudes as high as 5 km a.s.l., while over the Bay of Bengal the monsoon effect is observed through the steep decrease of the dust climatological extinction coefficient values (Fig. 4s), as a result of the wet deposition of aerosols. Values consistently below 25 Mm$^{-1}$ are observed over the Indonesia region throughout the year.

### 3.3 Conditional dust extinction coefficient

In this section we present and discuss the intensity of the dust events and the purity of dust aerosols in the atmosphere over SE Asia (three-dimensional). In order to investigate the intensity of the dust events over the domain of SE Asia the dust conditional extinction coefficient parameter is used, as defined in Sect 2. The vertical distributions of the dust conditional extinction coefficient and the corresponding conditional depolarization ratio are presented in Figures 5 and 6 respectively. More specific, Figure 5 shows the seasonal vertical distribution of the dust conditional extinction coefficient (Mm$^{-1}$) for nine years of

CALIPSO observations (01/2007-12/2015) and for the five zones of 10° latitudinal interval between 5° and 55° N. The vertical structure of the atmosphere is shown for altitudes higher than the average surface elevation of the CALIPSO orbits during the 9 year period between 01/2007 and 12/2015. The continuous and dashed lines correspond to the average elevation of the surface level and to the average maximum elevation of the surface elevation respectively.

Distinct sources of dust generation, where dust conditional extinction coefficient values exceed 200 Mm$^{-1}$ are revealed. High

values of dust conditional extinction coefficients indicate that Taklimakan and Gobi deserts are the most dominant sources of dust aerosols to the north of the Tibetan Plateau. To the east of the orographic barrier of Himalayas the major source of dust generation is the Thar Desert. In addition to the natural sources, regions of dust emissions related to anthropogenic activities are also evident. As seen in Figure 5, values that exceed 100 Mm$^{-1}$ are observed throughout the year over the highly-industrialized and densely-populated regions of SE China and over the Indian subcontinent. At the northern part of China

though, the near surface dust emissions to the west of the Tarim Basin most probably represent a mixture of Gobi and anthropogenic dust emissions. These features are consistent with the observation that close to the sources of dust generation the conditional extinction coefficient values are of the same magnitude as the climatological coefficient values.

Although the spatial and seasonal features between the observed conditional and climatological values are highly consistent, two major differences are evident: (1) the climatological values become significantly lower than the conditional values with

increasing distance from the sources of dust; and (2) the conditional values observed in the upper troposphere are significantly higher than the climatological values. The differences are attributed to the difference between the definitions of the dust conditional and climatological products. The dust climatological extinction product is related to the contribution of the dust load to the total aerosol load. By contrast, the dust conditional coefficient product describes exclusively the dust events. As a consequence, areas of rare dust events in general yield low climatological extinction coefficient values. This makes the

conditional coefficient values an ideal parameter in order to realistically describe and study the routes of transport of the dust plumes.



To the north and east of the Tibetan Plateau, as seen in Figure 5, two distinct eastward pathways of dust transport are evident: (1) a northern flow that propagates towards the Yellow Sea and the Pacific Ocean (Uno et al., 2009) and (2) a southern flow that occurs over central China (Kuhlmann and Quaas, 2010). Both transport pathways are observed over the middle and upper troposphere and their intensity is subject to high spatial and seasonal variability. Although the dust transports diverge towards

different directions, over the belt between 25° and 45° N dust conditional coefficient values that exceed 25 Mm⁻¹ are detected as high as 9 to 11 km altitude. While the distance from the Taklimakan and Gobi deserts increases, towards and over the Pacific Ocean, a negative gradient is observed. The maximum altitude where dust aerosols are observed during dust events decreases to less than 6 km a.s.l. over the regions both to the north (Mongolia and Manchuria Plains) and to the south of China (Indochina and Indonesia).

To the south and west of the Tibetan Plateau dust transport that originates from the Arabian Peninsula and Thar Desert and propagates towards the Indian Subcontinent (Gautam et al., 2009) and the Indian Ocean is observed. The maximum altitude and intensity of the flow of dust aerosols originating from the northwest part of India is subject to high seasonal oscillation. During the period between May and August dust events yield values of dust extinction coefficient as high as 200 Mm⁻¹ over the source of Thar Desert. The layer of dust over the Indian subcontinent during this period exceeds the altitude of 5 km a.s.l.,

while during the period between September and May the dust aerosols are constrained lower than 4 km a.s.l. Over Indonesia, Indochina and the Bay of Bengal, the dust aerosol layer is well-confined within the first 4 km a.s.l. throughout the year, with dust extinction coefficient values up to 30 Mm⁻¹ for heights greater than 1 km a.s.l. In the first km a.s.l. we see relatively height values that regularly exceed 40 Mm⁻¹. These values are affected from the selection of the particle depolarization ratio of the non-dust aerosols in our dust-separation methodology (as discussed in Section 3). In this technic we selected the most dominant

value for the depolarization of the non-dust aerosols (optimal for anthropogenic and marine cases). Furthermore under specific conditions the particle depolarization ratio of dry marine aerosols can exceed these values reaching up to 0.1, especially close to the top of the marine boundary layer (Haarig et al., 2017). By using the generic non-dust depolarization of 0.03, we have to recognize a bias in the marine boundary layer, extinction values up to 50% of the mean values of the conditional dust product. The depolarization ratio is an ideal intensive parameter for the discrimination between spherical and non-spherical aerosols,

hence for the classification of dust aerosols (Omar et al., 2009). Values of depolarization ratio at 532nm that exceed 30% denote the presence of pure dust aerosols (Liu et al., 2008c), while lower values that range between 10% and 30% suggest a mixture of dust with more spherical aerosols (Murayama et al., 2003; Tesche et al., 2009). Therefore, the depolarization ratio is used here as an indicator in order to describe the state of the dust mixture and as a discriminator between pure dust and polluted dust cases.

Figure 6 shows the vertical, horizontal and seasonal variability of dust depolarization ratio for nine years of CALIPSO observations (01/2007-12/2015) and for five zones of 10° latitudinal interval, between 5° and 55° N. The vertical cross sections of the mean depolarization ratio correspond to the dusty CALIPSO observations (dust and polluted dust cases), hence correspond to the dust conditional extinction coefficient parameter described above (Fig.5). Based on Figure 6, dust depolarization ratio values between 30% and 35% are regularly observed over the Taklimakan, Gobi and Thar deserts

throughout the year. Intermediate depolarization ratio values, between 25% and 35%, are observed close to the dust sources, while even lower values, between 10% and 25%, are evident over the densely-populated and highly-industrialized regions of SE China and India and over the remote domains of Indonesia and Indochina.

In general, to the north of Himalayas (Fig. 6e-h), a structure of three different height ranges is evident. The low dust depolarization ratio values observed over the densely-populated and highly-industrialized regions suggest the occurrence of a

mixture of non-spherical aerosols with particles of anthropogenic origin (Heese and Wiegner, 2008). Conversely to the aerosol layers close to the urban/industrial regions, the elevated layers over China and India are characterized by intermediate dust depolarization ratio values, from about 15% to 25%. The higher values of dust depolarization ratio in the middle/upper troposphere compared to the lower troposphere are consistent with the characteristics of dust transport. More specifically, to





the north of the Tibetan Plateau, between 25º and 45º N, three ranges of dust depolarization ratio are observed. The air masses below 2 km altitude are characterized by significantly low dust depolarization ratio, values in general below 15%. The observed low values are most probably the effect of anthropogenic emissions coupled with near-surface dust aerosols. The altitudinal range between 2 and 4 km height is characterized in general by depolarization ratio values greater than 15%, which though

rarely exceed the value of 20%. The third elevated layer, above about 4 km, is characterized by depolarization ratio values greater than 20%. The dust layers between 2 and 4 km height and above 4 km height have been observed and identified as dust aerosol layers with different origin, from Gobi and Taklimakan deserts respectively (Kwon et al., 1997; Matsuki et al., 2003). To the west of the Tibetan Plateau, between 25º and 35º N, Thar Desert is located. Over Thar Desert dust aerosols yield dust depolarized values greater than 25% throughout the year. Dust depolarization values that exceed 30% are observed during JJA,

when Thar dust activity is at its maximum (Fig. 6k). The elevated layer of dust is accordingly transported eastwards, over the highly-industrialized and dense-populated Indo-Gangetic plains. The interaction of dust aerosols with locally generated aerosol particles (Middleton, 1986) is evident through the decrease of the dust depolarization ratio over the Indian subcontinent. The observed depolarization ratio values range between 15% and 20% over the Indo-Gangetic plains and the foothills of Himalayas (Fig. 6i-l). Furthermore, dust depolarization ratio values observed to the west of the Himalayas are typically larger than the

values observed over the eastern Himalayas and over the Tibetan Plateau. The observed intermediate values of dust depolarization ratio values at the windward slopes of the Himalayas are consistent with the Elevated Heat Pump hypothesis (Lau et al., 2006), which assess the accelerating Himalayas warming as the effect of accumulation of dust aerosols coupled with black carbon over the Tibetan Plateau.

The dust depolarization ratio values over Indochina and Indonesia are significantly different from the corresponding

depolarization ratio features observed over China and India (Fig. 6m-t). The dust depolarization values to the south of the Tibetan Plateau and to the east of the Indian subcontinent are in general below 15% indicating that the dust aerosols are coupled with natural and anthropogenic emissions.

### 3.4 Temporal evolution of AOD and D_AOD

In this section the CALIPSO AOD and D_AOD timeseries, based on nine years of overpasses, are presented and discussed. In

addition to the CALIPSO/CALIOP AOD and D_AOD trends, Aqua/MODIS AOD trends for the same period (01/2007-12/2015) are presented. The short-term trends in this paper are calculated through the method originally proposed by Weatherhead et al. (1998). The applied method has been broadly used to examine the trends of trace gasses, aerosols and surface solar radiation (e.g. De Smedt et al., 2010; de Meij et al., 2012; Pozzer et al., 2015; Georgoulias et al., 2016; Alexandri et al., 2017). Monthly satellite-based time series are fitted by using a model with a linear trend and a Fourier-based seasonal

component for the annual cycle. According to the method, the calculated trend ($\omega$) is statistically significant at the 95% confidence level if the absolute value of the ratio of $\omega$ to its precision ($\sigma_\omega$) is greater than 2 ($|\omega/\sigma_\omega|>2$). The approach followed here is extensively described in Alexandri et al. (2017).

Figure 7 shows the short-term trends of CALIOP/CALIPSO and MODIS/Aqua over SE Asia for the period 01/2007-12/2015. MODIS/Aqua $AOD_{550}$ trends were calculated from the C6 DTDB merged $AOD_{550}$ dataset and are presented in Fig.7b. Trends

and their statistical significance at the 95% confidence level were computed based on the Weatherhead et al. (1998) methodology. For computing the CALIOP/CALIPSO $AOD_{532}$ trends the methodology includes additionally a spatial expansion of each grid, in order to increase the accuracy and representativity of each $AOD_{532}$ value in the sequence of the monthly mean CALIOP/CALIPSO time series. The mean optical depth value per month is computed based not only on each grid but additionally on the corresponding eight surrounding neighbour grids. Finally the methodology proposed by

Weatherhead et al. (1998) is applied to examine the statistical significance of CALIOP/CALIPSO $AOD_{532}$ (Fig.7a). On the trend plots the "+" symbol denotes trends statistically significant at the 95% confidence level. Negative trends are shown in





blue, while red colour indicates positive trends. In addition to the short-term trends the mean Aqua Cloud Fraction (Fig.7d) and the number of months used in CALIPSO time series (Fig.7c) are shown.

Regarding China, CALIOP shows significantly positive $AOD_{532}$ trends over the northwest and eastern provinces whereas negative statistically significant trends are mostly found over the southeastern provinces. MODIS shows statistically significant

positive $AOD_{550}$ trends over northwest, central and eastern China; where over northeast China AOD trends are mostly positive. More specifically, both CALIOP and MODIS sensors quantitatively agree on a statistically significant increase at the 95% confidence level over Xinjiang (0.007 $yr^{-1}$) and Hebei (0.01 $yr^{-1}$) provinces. Towards central China and the Tibetan Plateau differences are observed between CALIOP and MODIS. Over the broader Tibetan Plateau low positive trends are shown by MODIS while no trends is found by CALIOP. However the disagreement over the region of Tibetan Plateau (Tibet, Qinghai,

Gansu and Yunnan provinces) is not relevant considering both the absence of statistical significance and the small magnitude of the AOD trends. Larger differences between the MODIS and CALIOP short-term trends are detected over the SE China. MODIS shows a strong significant increase of $AOD_{550}$ (0.01 $yr^{-1}$) whereas significantly decreasing $AOD_{532}$ trends are visible by CALIOP (-0.007 $yr^{-1}$). Over the Indian Peninsula MODIS and CALIOP trends qualitatively agree over the 9-year period in most regions. Positive AOD trends (0.01 $yr^{-1}$) are found by both sensors over the broader central and eastern Indian Peninsula,

although disagreements are observed over the western regions of India. The AOD trends of CALIOP and MODIS are very similar over land, while over ocean (Arabian Sea, Bay of Bengal), discrepancies are observed. Strong statistical increase is observed by MODIS over the Arabian Sea (0.01 $yr^{-1}$), while not statistical significant positive $AOD_{550}$ trends are present over the Bay of Bengal (0.002 $yr^{-1}$). The strongly increasing $AOD_{532}$ trend over the Arabian Sea though is not corroborated by CALIOP observations. The observed trends from CALIOP and MODIS should be interpreted and compared with caution,

since the observed discrepancies between MODIS and CALIOP AOD short-term trends may be attributed to several aspects such as the different measurement principles and sampling of the two sensors among others. Differences between the two sensors have been reported in the literature (Redemann et al., 2012).

Three domains of interest are selected to perform regional analysis on the CALIPSO AOD and D_AOD time series: Southeastern Asia (5°-55°N, 65°-155°E), Eastern China (22°-43°N, 105°-124°E) and South Asia (5°-33°N, 65°-91°E). Figure 8

shows the linear trends of D_AOD (left column) and AOD (right column) for the selected domains. The continuous black lines represent the seasonal variability between 01/2007 and 12/2015, while the dashed lines depict statistical significant (non-statistical significant) trends. The scatter points in grey denote the monthly mean D_AOD and AOD values over the selected areas, while estimated trends (1/yr) and changes of D_AOD and AOD with respect to 2007 (%/yr) are additionally shown.

Focusing on the selected domains, over Southeastern Asia (AS) the D_AOD trend is negative (-5x10$^{-4}$ $yr^{-1}$ / 0.96% $yr^{-1}$), while

despite the decreasing D_AOD trend over the AS domain, the AOD trend is slightly positive (4x10$^{-4}$ $yr^{-1}$ / 0.21% $yr^{-1}$). Both the D_AOD and AOD trends over AS though are not statistically significant. In contrast to the AS region, the observed AOD trend over Eastern China (EC) is statistically significant negative (-5x10$^{-3}$ $yr^{-1}$ / -1.38% $yr^{-1}$). Similar to the AOD, the region of EC presents a statistically significant negative D_AOD trend (-2.1x10$^{-3}$ $yr^{-1}$ / -2.28% $yr^{-1}$). The negative AOD trend observed over EC is in line with the air quality regulations and the applied policies promoting the reduction of emissions over China.

Van der A et al. (2017) report on the decrease of aerosol precursor gases ($SO_2$, $NO_2$) based on OMI, SCIAMACHY and GOME-2 satellite observations, while Yoon and Pozzer (2014) report on the decrease of biomass-burning related emissions. The decline of AOD over China is additionally observed by several authors (Kang et al., 2016, Zhang et al., 2017), with the suggested pivot point around 2011 (Zhao et al., 2017). The negative AOD trend over Southeastern China is further enhanced due to the negative D_AOD trend, which in turn can be attributed to the positive precipitation trend over Southeastern China

and the increase of dust aerosol deposition (Pozzer et al., 2015). Focusing over the South Asia (SA) region though similar behaviour as over AS is observed. SA is characterized by not statistically significant decreasing D_AOD trends (-1.5x10$^{-3}$ $yr^{-1}$ / -1.01% $yr^{-1}$). In contrast, due to the increasing emissions of the fast-developing India, the observed AOD trend is positive (3.3x10$^{-3}$ $yr^{-1}$ / 0.98% $yr^{-1}$, not significant).





#### 4 Summary and conclusions

In this work, CALIPSO is used to provide a multiyear 4-D climatology of desert dust aerosols over South-East Asia. An optimized dust aerosol product, developed using CALIOP backscatter and particle depolarization ratio, along with a regional correction on dust lidar ratio suitable for Asian dust is used. The optimized product is utilized to provide the horizontal and

vertical distributions and the temporal evolution of dust aerosols over a 9-year period (01/2007-12/2015) at a spatial resolution of 1 deg.

Our analysis shows similar patterns in the horizontal distribution of AOD, D_AOD and non-dust AOD between all four seasons, although the magnitude of the observed patterns varies with season. High values of non-dust AOD are consistently observed over the heavily industrialized and densely populated regions of China and India. Regarding the horizontal

distribution of dust aerosols the source regions of Taklimakan, Gobi and Thar deserts are clearly mapped through the systematic high D_AOD values throughout the year, although the magnitude of the observed D_AOD features is subject to high seasonality. Maximum activity of Gobi and Taklimakan deserts is observed during MAM, while Thar Desert's highest activity is mostly evident during JJA. The seasonality of the dust transport pathways is additionally well-captured. Dust transport over the Indian Peninsula is more prominent during MAM and JJA, while over China similar patterns of a persistent dust aerosol

background is evident throughout the year, with a peak during MAM when the dust transport across the Pacific Ocean is more pronounced. Regarding the vertical distribution of dust aerosols, the utilized dust CoM, TH and the dust extinction coefficient corroborate the vertical structure and transport pathways of dust aerosols over SE Asia. Furthermore, through the extinction coefficient profiles our study quantitatively addresses the vertical climatological contribution of dust aerosols to the total aerosol load and the intensity of the dust events, on a seasonal basis. Regarding the temporal evolution of AOD and D_AOD

our analysis indicates statistically significant positive short-term AOD trends over the Indian Peninsula, NW China and E China, whereas our study shows negative short-term AOD trends over SE China. CALIPSO AOD trends are generally in qualitative agreement with the derived MODIS AOD trends over large domains of SE Asia, although the short-term trends disagree over specific regions. The CALIOP and MODIS trends though are interpreted and compared with caution, since the samples of the datasets are non-uniform.

Observational evidence regarding the vertical distribution of dust layers is of particular interest for modelling studies and consequently for assessing the role of airborne dust on radiation (direct climate effect) and clouds (indirect climate effect). So far, modelling simulations of dust are evaluated through comparisons with column dust observations (e.g. AERONET and MODIS AOD) and only occasionally with lidar or in-situ measurements at specific stations. Moreover, the availability of ground measurements (lidar and in-situ) is limited near the dust sources. Assimilation of dust in atmospheric models is also

problematic since the 2D initial observational fields need to be assimilated towards the 3D prognostic model variables. Other than AOD aerosol properties (e.g. Ångström exponent or aerosol type) are essential for radiative forcing studies as the spectral dependence of AOD impacts the RF model results for different aerosol cases (e.g. dust or non-dust cases). Furthermore, studies of transboundary aerosol transport (e.g. China, Korea and Japan) should include a quantification of natural (dust) and anthropogenic aerosol components. In this context, utilization of the CALIPSO pure dust profiles derived here will certainly

assist both the evaluation and assimilation activities in relevant atmospheric simulations and will provide better estimation of the climatic impact of dust aerosols.

#### Acknowledgements

The authors acknowledge support through the following projects and research programs:

MarcoPolo under grant agreement n° 606953 from the European Union Seventh Framework Programme (FP7/2007-2013).
ESA-ESTEC project LIVAS (contract N°4000104106/11/NL/FF/fk).





BEYOND under grant agreement no. 316210 of the European Union Seventh Framework Programme: FP7-REGPOT-2012-2013-1.

ACTRIS-2 under grant agreement no. 654109 from the European Union's Horizon 2020 research and innovation programme ECARS under grant agreement No 602014 from the European Union's Horizon 2020 Research and Innovation programme.

European Research Council under the European Community's Horizon 2020 research and innovation framework program / ERC Grant Agreement 725698 (D-TECT).

The authors acknowledge EARLINET for providing aerosol lidar profiles available under the World Data Center for Climate (WDCC). We thank the AERONET PIs and their staff for establishing and maintaining the AERONET sites used in this investigation. CALIPSO data were obtained from the ICARE Data Center (http://www.icare.univ-lille1.fr/). CALIPSO data

were provided by NASA. We thank the ICARE Data and Services Center for providing access to the data used in this study and their computational center.

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



## 5    Tables and Figures

**Figure 1.** Illustration of the under study domain,  confined between longitudes 65°-155° E and latitudes 5°-55° N. Major dust aerosol sources (yellow colour) are included (Taklimakan, Gobi, Thar deserts). Dashed black lines delineate domains of high surface elevation (Tibetan Plateau, Himalayan Ridge). The grey lines delineate the domains of regional statistics provided in table 1: (1) Taklimakan / Gobi Deserts, (2) Tibetan Plateau, (3) SE China, (4) Indian Peninsula, (5) North Pacific Ocean, (6) Indochina / Indonesia.

**Figure 2:** Spatial distribution of the seasonal mean CALIPSO Aerosol Optical Depth (AOD), Dust Optical Depth (D_AOD), Optical Depth difference between AOD and D_AOD (ΔOD) and of the percentage of D_AOD with respect to the total AOD for the domain between 65°-155° E and 5°-55° N and for the period 01/2007-12/2015.

**Figure 3:** Spatial distribution of dust occurrence [%], dust CoM (Center of Mass) and dust TH (Top Height) in km a.s.e., for each season over the domain between 65°-155° E and 5°-55° N and for the period 01/2007-12/2015.

**Figure 4:** Geographical zonal distribution of the climatological dust extinction coefficient (Mm$^{-1}$) profiles for the regions with longitude from 65° to 155° E and latitudes 45° to 55° N (a-d), 35° to 45° N (e-h), 25° to 35° N (i-l), 15° to 25° N (m-p), 15° to 25° N (m-p), 5° to 15° N (q-t), as illustrated by the domain maps on the left; profiles are presented as three-month averages: December-January-February (a,e,i,m,q), March-April-May (b,f,j,n,r), June-July-August (c,j,k,o,s) and September-October-November (d,h,l,p,t). The minimum terrain elevation is denoted with black colour. The black continuous (dashed) line refers to the mean (max) elevation of the surface.

**Figure 5:** Geographical zonal distribution of the conditional dust extinction coefficient (Mm$^{-1}$) profiles for the regions with longitude from 65° to 155° E and latitudes 45° to 55° N (a-d), 35° to 45° N (e-h), 25° to 35° N (i-l), 15° to 25° N (m-p), 15° to 25° N (m-p), 5° to 15° N (q-t), as illustrated by the domain maps on the left; profiles are presented as three-month averages: December-January-February (a,e,i,m,q), March-April-May (b,f,j,n,r), June-July-August (c,j,k,o,s) and September-October-November (d,h,l,p,t). The minimum terrain elevation is denoted with black colour. The black continuous (dashed) line refers to the mean (max) elevation of the surface.

**Figure 6:** Geographical zonal distribution of the conditional dust depolarization ratio profiles for the regions with longitude from 65° to 155° E and latitudes 45° to 55° N (a-d), 35° to 45° N (e-h), 25° to 35° N (i-l), 15° to 25° N (m-p), 15° to 25° N (m-p), 5° to 15° N (q-t), ας illustrated by domain maps on the left; profiles are presented as three-month averages: December-January-February (a,e,i,m,q), March-April-May (b,f,j,n,r), June-July-August (c,j,k,o,s) and September-October-November (d,h,l,p,t). The minimum terrain elevation is denoted with black colour. The black continuous (dashed) line refers to the mean (max) elevation of the surface.

**Figure 7:** Trends of MODIS/Aqua and CALIOP/CALIPSO AOD532nm over SE Asia for the period 01/2007-12/2015. Left column: CALIOP/CALIPSO trends of AOD532nm (a) and the number of months used in the CALIOP/CALIPSO time series (c). Right column: MODIS/Aqua C6 AOD550nm trends of DTDB merged datasets (b) and mean Cloud Fraction (d).  Symbol "+" denotes trends statistically significant at the 95% confidence level.

**Figure 8:** Short-term time series of CALIPSO/CALIOP AOD and D_AOD over SE Asia (AS), of Eastern China (EC) and of South Asia (SA) based on observations during the period 01/2007-12/2015. The dashed lines show the multi-annual trend line while the continuous black lines depict the seasonal variability.



**Table 1: Domain statistics on mean Dust Optical Depth, max D_AOD / 95th percentile, dust Center of Mass (CoM) and Top Height (TH) (both in km a.s.e.) and number of dust profiles to the total number of cloud-free profiles, based on the period 01/2007-12/2015.**

| | Mean D_AOD ± SD (Climatological) | D_AOD Max / Percentile 95% (Climatological) | Dust CoM ± SD (km / a.s.e.) (Conditional) | Dust Top Height ± SD (km / a.s.e.) (Conditional) | Nr of Dust in Nr of cloud-free |
|---|---|---|---|---|---|
| **Taklimakan / Gobi** | | | | | |
| DJF | 0.078 ± 0.135 | 1.802 / 0.327 | 2.31 ± 1.39 | 3.47 ± 2.01 | 0.74 |
| MAM | 0.193 ± 0.308 | 2.729 / 0.819 | 3.06 ± 1.43 | 5.01 ± 2.17 | 0.78 |
| JJA | 0.113 ± 0.232 | 2.504 / 0.529 | 3.19 ± 1.38 | 4.94 ± 1.8 | 0.68 |
| SON | 0.095 ± 0.18 | 2.488 / 0.401 | 2.58 ± 1.27 | 3.92 ± 1.77 | 0.73 |
| **Tibetan Plateau** | | | | | |
| DJF | 0.012 ± 0.037 | 0.758 / 0.062 | 6.01 ± 1.26 | 6.96 ± 1.42 | 0.31 |
| MAM | 0.028 ± 0.055 | 0.731 / 0.127 | 6.2 ± 1.21 | 7.76 ± 1.64 | 0.52 |
| JJA | 0.013 ± 0.033 | 0.676 / 0.068 | 5.99 ± 1.05 | 7.13 ± 1.3 | 0.39 |
| SON | 0.006 ± 0.023 | 0.631 / 0.032 | 6.08 ± 1.37 | 6.87 ± 1.45 | 0.24 |
| **SE China** | | | | | |
| DJF | 0.062 ± 0.104 | 1.769 / 0.254 | 1.73 ± 1.29 | 3.03 ± 1.84 | 0.79 |
| MAM | 0.108 ± 0.171 | 2.39 / 0.404 | 2.38 ± 1.47 | 4.35 ± 2.26 | 0.85 |
| JJA | 0.032 ± 0.064 | 1.06 / 0.145 | 1.9 ± 1.56 | 3.09 ± 2.13 | 0.69 |
| SON | 0.045 ± 0.081 | 1.329 / 0.19 | 1.6 ± 1.27 | 2.74 ± 1.71 | 0.72 |
| **Indian Peninsula** | | | | | |
| DJF | 0.043 ± 0.065 | 1.736 / 0.147 | 1.13 ± 0.9 | 2.13 ± 1.28 | 0.84 |
| MAM | 0.171 ± 0.188 | 1.944 / 0.521 | 1.79 ± 0.99 | 3.63 ± 1.44 | 0.93 |
| JJA | 0.199 ± 0.167 | 2.071 / 0.751 | 2.05 ± 1.22 | 3.72 ± 1.55 | 0.86 |
| SON | 0.075 ± 0.106 | 1.459 / 0.267 | 1.29 ± 0.87 | 2.54 ± 1.25 | 0.83 |
| **N Pacific** | | | | | |
| DJF | 0.026 ± 0.059 | 1.169 / 0.113 | 1.55 ± 1.54 | 2.49 ± 1.97 | 0.67 |
| MAM | 0.046 ± 0.085 | 1.596 / 0.196 | 2.29 ± 1.86 | 3.96 ± 2.7 | 0.79 |
| JJA | 0.007 ± 0.02 | 0.613 / 0.035 | 1.55 ± 1.95 | 2.29 ± 2.38 | 0.45 |
| SON | 0.012 ± 0.032 | 0.929 / 0.057 | 1.26 ± 1.49 | 2.06 ± 1.89 | 0.56 |
| **Indochina / Indonesia** | | | | | |
| DJF | 0.005 ± 0.016 | 0.562 / 0.022 | 0.84 ± 0.99 | 1.28 ± 1.11 | 0.39 |
| MAM | 0.005 ± 0.012 | 0.269 / 0.024 | 0.98 ± 1.16 | 1.47 ± 1.29 | 0.47 |
| JJA | 0.003 ± 0.01 | 0.383 / 0.018 | 1.19 ± 1.96 | 1.6 ± 2.07 | 0.33 |
| SON | 0.003 ± 0.012 | 0.712 / 0.016 | 1.02 ± 1.57 | 1.45 ± 1.71 | 0.38 |



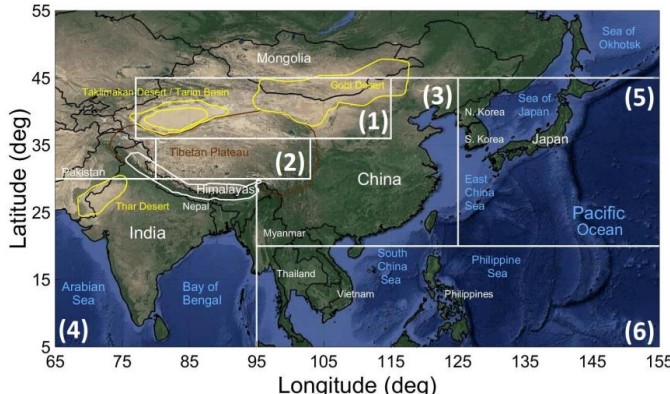

**Fig.1**



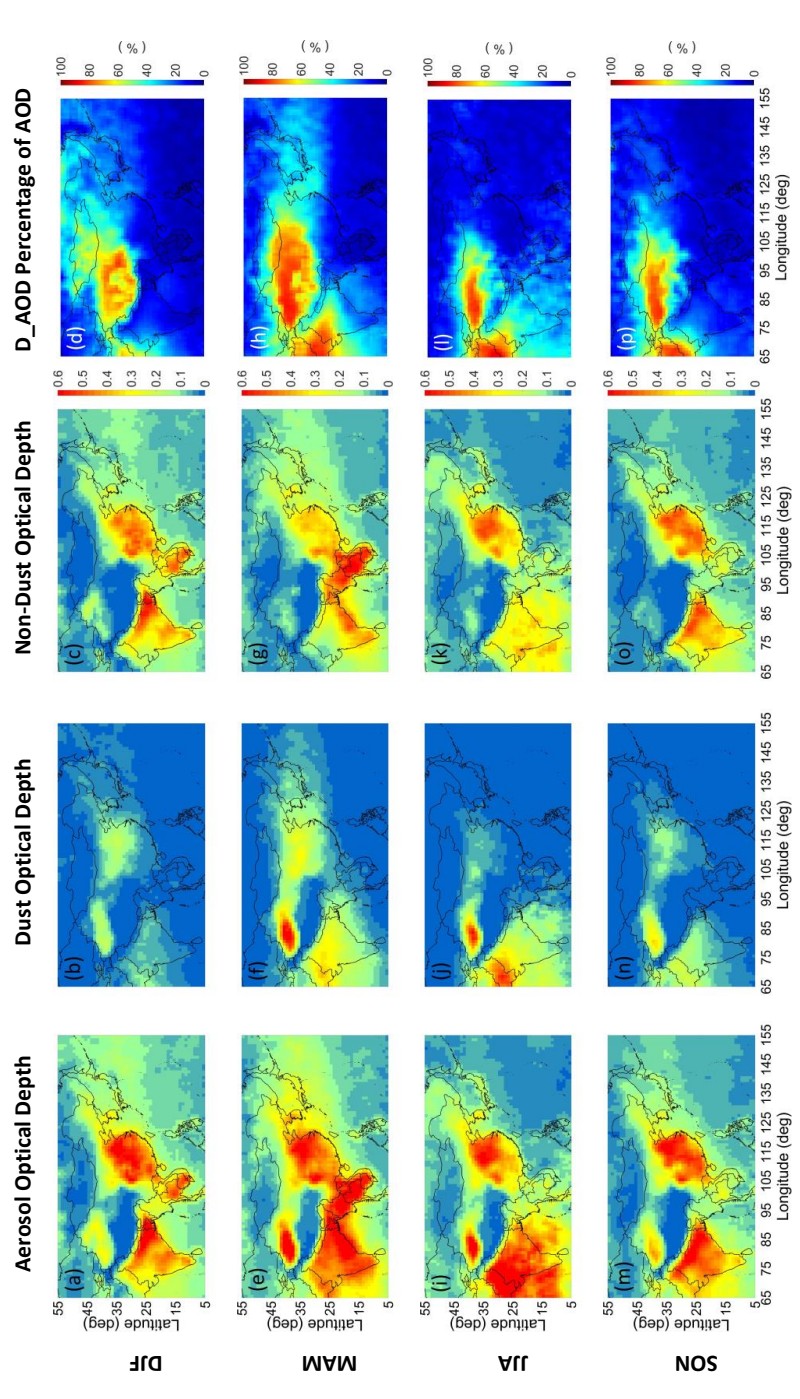

Fig.2



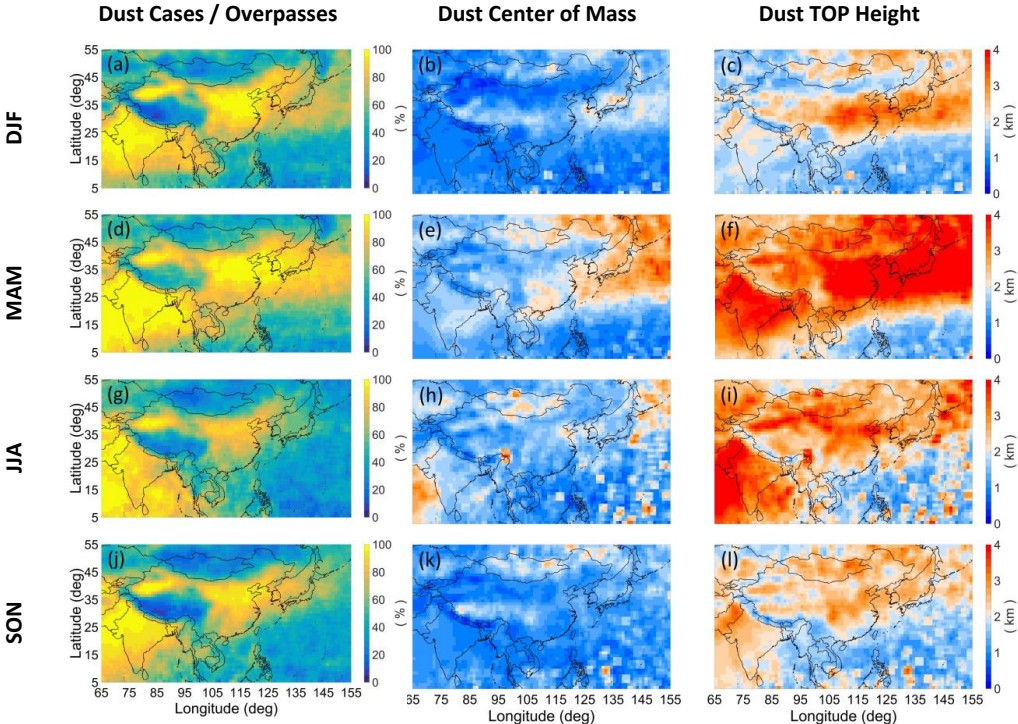

5    **Fig. 3**





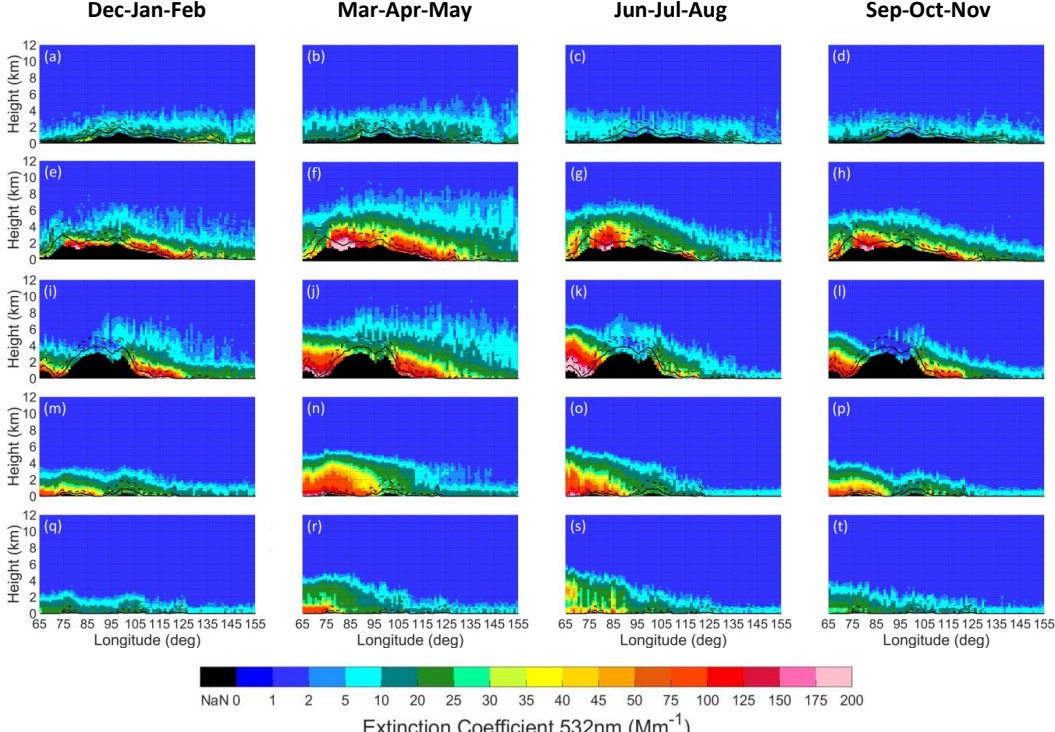

**Fig. 4**





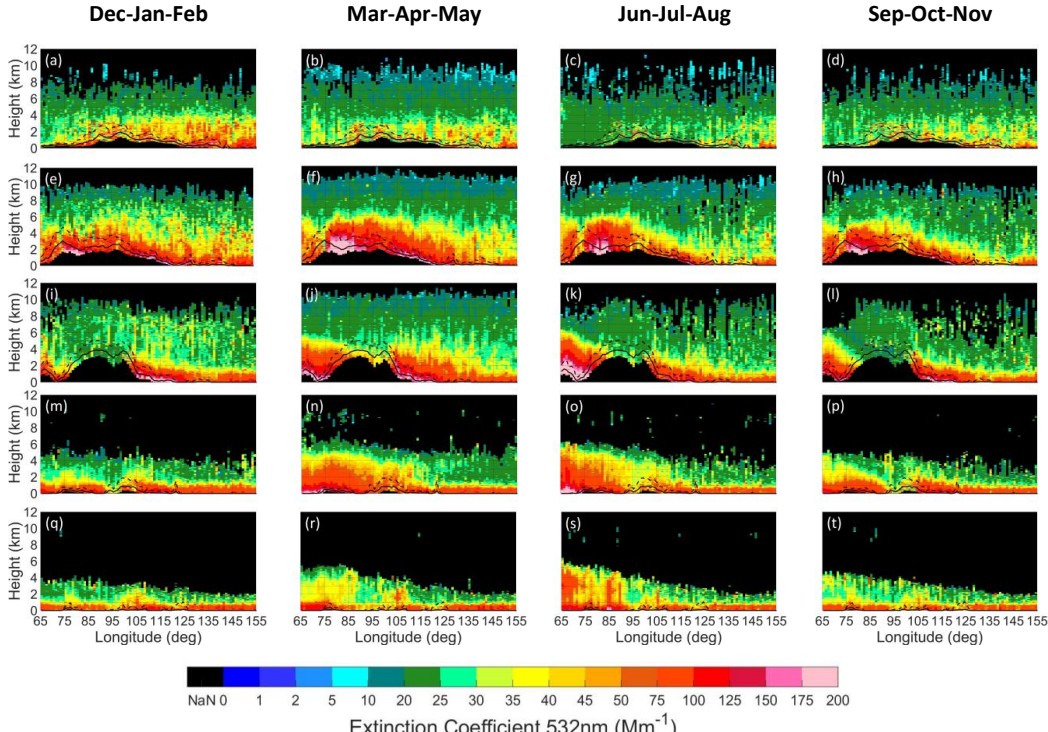

**Fig. 5**



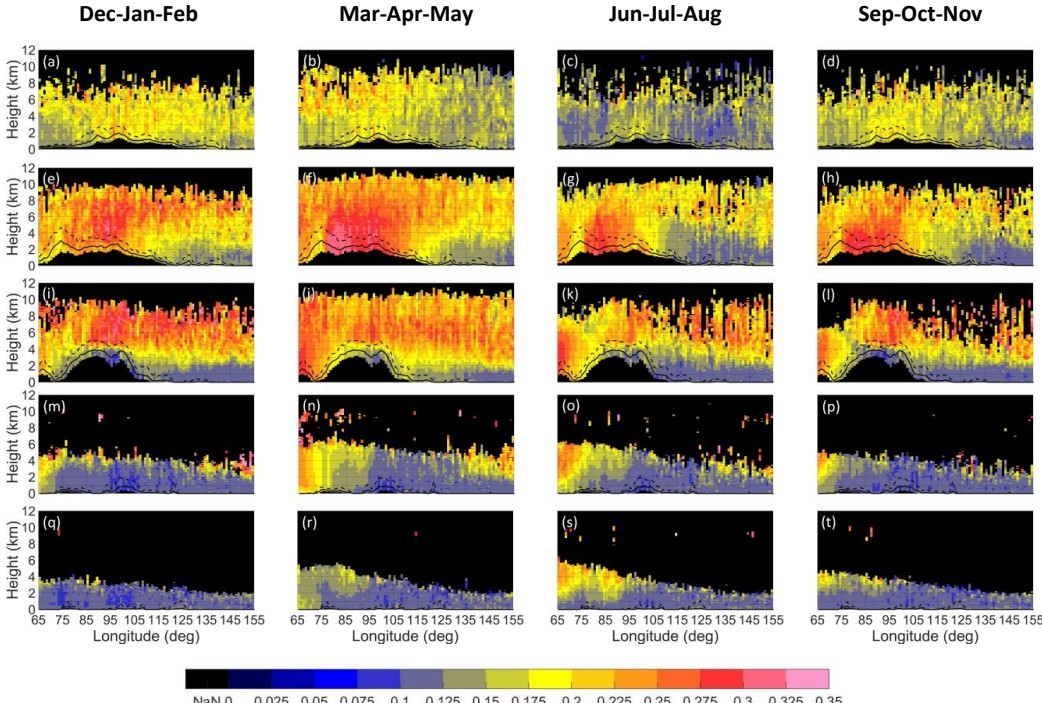

**Fig. 6**



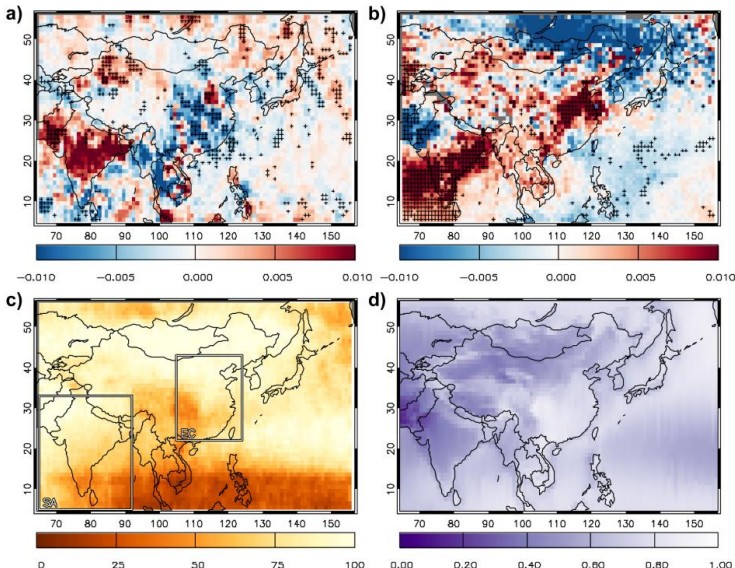

**Fig. 7**





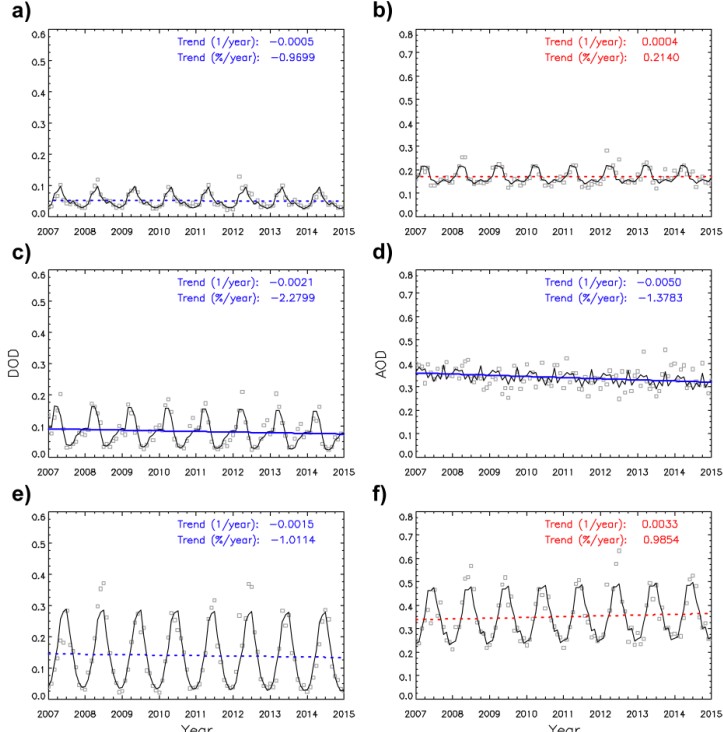

Fig. 8