# Peer review of "9-year spatial and temporal evolution of desert dust aerosols over South and East Asia as revealed by CALIOP"

_Atmospheric Chemistry and Physics, 2017_

## Referee Comment (RC1) · Anonymous Referee #1 · 3 Oct 2017

Comments to the Author (General comments) In this paper, the spatial and temporal evolution of desert dust aerosols over South-East Asia has been systematically investigated based on CALIPSO since it can provide much information about aerosols. However, I'm interesting to see this paper published before revised as below suggestion. (Specific comments) 1. Page1, Line 1, "Dust aerosols have a significant role on climate through the direct radiative effect of absorption and scattering of solar and thermal terrestrial radiation ". I think you should add the reference : ïĄň Huang J P , Fu Q , Su J , T ang Q , Minnis P , Hu Y , Y i Y , Zhao Q . 2009 . T aklimakan dust aerosol radiative heating derived from CALIPSO observations using the Fu - Liou radiation model with CERES constraints. Atmos Chem Phys , 9: 401 1 –4021 ïĄň Chen,

[Figure]

S., J Huang,J. Li,R. Jia, N. Jiang, L. Kang, X. Ma, T. Xie, Comparison of dust emission, transport, and deposition between the Taklimakan Desert and Gobi Desert from 2007 to 2011, 2017, Science China Earth Sciences, 60:1-1, doi:1.01007/s11430-016-9051-0. 2. In the paragraph 1, the semi-effect of dust should be also added. The effect can be seen from the references: ïĄň Huang J P , Lin B , Minnis P , W ang T , W ang X , Hu Y , Y i Y , A yers J K . 2006a . Satellite - based assessment of possible dust aerosols semi - direct effect on cloud water path over East Asia. Geophys Res Lett , 33:L19802 3. Line10, "airborne mineral dust is considered a significant atmospheric aerosol contributor", should be corrected into ". . .considered as a. . ." 4. Page2, Line29-32," Although passive satellite sensors provide information on the column properties of aerosols with adequate spatial and temporal resolution, they are bound to certain limitations, the major limitation being the lack of information on the three- 30 dimensional distribution (vertical profile) of aerosols in the atmosphere, an important information for the assessment of the aerosols radiative forcing on climate as well as their contribution as IN and CCN (IPCC 2013)." This sentence is too long too understand it means, please rewrite it. 5. Page4, line10 ,I think you should delete this words :"n order to discriminate the detected atmospheric features types into subtypes", because we have known the goal of the classification algorithm before this sentence. 6. Page4, line 14, the "2ox5o grid resolution" should be corrected into "2° x 5°". 7. Page4, line17, "1ox1o" need to be corrected. the whole paper should be checked again 8. Page6, line18, since you have said that the daytime minimum and nighttime minimum, what dose the "minimum detectable AOD of 0.005" mean? 9. Page4, Line24, what dose the "SAMUM" mean? Please write the full name. 10. Page4, line51, in this paragraph you introduce the methods of distinguish pure dust and non dust. However, I still don't know the differences of the CALIPSO product of dust and polluted dust with the pure dust and non dust. Since we can directly derive the dust extinction coefficient and profiles from the product, why don't you use it? And what about merits of the method to select the pure dust? What's the differences of the pure dust and dust products directly from CALIPSO L2? 11. Page5,line 30, please check this sentence of "The seasonal zonal

ACPD
distribution of the climatological and conditional dust extinction coefficient (Mm-1)". If it's right to explain it. 12. Page5, line30, I want to know weather the climatological dust extinction coefficient means the aerosol extinction without dust extinction coefficient since you write this sentence "This is accomplished by setting the dust extinction coefficient value of 0 km-1, for observations with non-dust aerosols". And the conditional dust product only has the dust extinction coefficient. 13. Page7, line33, what does the "N. China" mean? 14. Page8, line11, from the figure 3 ,the differences of dust frequency in the four seasons are not clearly, and the minimum in Fig. 3a is not obvious. 15. Page8,line39, please explain the pattern of the dust transport since you said "however, the pattern reverses (Fig. 3i)"
* * *

---

## Referee Comment (RC2) · Anonymous Referee #2 · 5 Dec 2017

The paper by Proestakis et al. present a 9-year climatology derived with mainly CALIOP measurements of the aerosol (dust) conditions over East and South Asia. The main focus is set on dust distribution but also non-dust aerosol is discussed. These novel results are thus of interest for atmospheric research and give a good overview of the dust distribution in this part of the world. Therefore in principle the paper is of interest for publication in ACP, however I recommend major revisions before it can be published. This is further explained below.

Major comments:

In my opinion the naming of the study area is misleading. Even though South-East Asia

may not be a protected phrase, many people have a different understanding concerning the region called this way. See for example:

https://en.wikipedia.org/wiki/Southeast_Asia

Therefore, I recommend to find a better name for the study area (e.g. South and East Asia or whatever) to avoid confusion and change the title and text accordingly.

The difference between the climatological and conditional dust product needs be more discussed in both, the methodology section ( I just understood the difference when reading Marinou 2017 but as this is essential it should be explained more explicitly here), but also in the result section. The reader is left alone with contradictory statements, like for example the dust top height which seems to be completely different between the two products. Therefore it should be clearly discussed:

(1) Which product can be used for which purpose,?

(2) Does it make sense to use this two different products, if yes why and for what?

(3) What we can learn from the two products presented here with respect to South and East Asia.

Conclusion: The current conclusion is not very informative. Thus, it should be really overworked to highlight new things and discuss what lessons have been learned, i.e. what are new results or newly gathered knowledge or does your study just confirm former studies etc...

Please check spelling and grammar intensively again. There are many sentences which are fractals, i.e. words are missing. Furtermore, many commas are missing etc...considering the bunch of co-workers this should be no problem.

Further specific comments are in the attached pdf.

Please also note the supplement to this comment:

https://www.atmos-chem-phys-discuss.net/acp-2017-797/acp-2017-797-RC2-supplement.pdf

[Figure]

**Supplement:**

[Figure]

**9-year spatial and temporal evolution of desert dust aerosols over**
[Figure]
South-East Asia as revealed by CALIOP

[revised manuscript text omitted]

**Number: 1**    Author: Referee    Date: 05.12.2017 13:22:38
doubled information, please cross check

**Number: 2**    Author: Referee    Date: 05.12.2017 13:22:38
which one?

**Number: 3**    Author: Referee    Date: 05.12.2017 13:27:43
table or flowchart or something showing what is climatological and conditional

**Number: 4**    Author: Referee    Date: 05.12.2017 13:22:38
what does filtered mean in this context?

**Number: 5**    Author: Referee    Date: 05.12.2017 13:22:38
aerosol mixture

**Number: 6**    Author: Referee    Date: 05.12.2017 13:22:38
Do you have some reference for that?

[revised manuscript text omitted]

**Number: 1**   Author: Referee   Subject: Comment on Text   Date: 05.12.2017 13:28:01

This has nothing to do with methodology right? At least it is not at the right place here.

**Number: 2**   Author: Referee   Subject: Comment on Text   Date: 05.12.2017 13:22:38

This is not South East Asia ;-)

**Number: 3**   Author: Referee   Subject: Comment on Text   Date: 05.12.2017 13:22:38

is this the pure dust bsc or ex. coeff.?

**Number: 4**   Author: Referee   Date: 05.12.2017 13:22:38

backscatter or extinction?

**Number: 5**   Author: Referee   Date: 05.12.2017 13:22:38

no sentence. And I guess here is something very important missing!

**Number: 6**   Author: Referee   Date: 05.12.2017 13:22:38

I am sorry, I do not understand the difference from what is written here!

**Number: 7**   Author: Referee   Date: 05.12.2017 13:22:38

I really do not understand the difference and how it is exactly defined. As this two products are essential for this publication it should be explained in more detail!

**Number: 8**   Author: Referee   Date: 05.12.2017 13:22:38

which domain?

**Number: 9**   Author: Referee   Date: 05.12.2017 13:22:38

which product?

**Number: 10**   Author: Referee   Date: 05.12.2017 13:22:38

no sentence

[revised manuscript text omitted]

35   dust generation and transport and the removal processes of aerosols from the atmosphere vary with season [5] the seasonal approach is selected. The seasons are defined as follows: December-January-February (DJF), March-April-May (MAM), June-July-August (JJA) and September-October-November (SON). Figure 2 shows the spatial distribution of the seasonal-mean AOD (Fig. 2a, e, i, m), D_AOD (Fig. 2b, f, j, n), [6] OD computed as the difference between the total AOD and D_AOD (Fig. 2c, g, k, o) and the corresponding percentage of D_AOD to the total AOD (Fig. 2d, h, l, p) in 1ox1o grid resolution and based

40   on 9 years of CALIPSO observations (01/2007-12/2015).

Number: 1      Author: Referee      Date: 05.12.2017 13:22:38

.

Number: 2      Author: Referee      Date: 05.12.2017 13:22:38

which wavelength?

Number: 3      Author: Referee      Date: 05.12.2017 13:22:38

Really????? The uncertainty of the AOD is more than 100%? If yes you need certainly to discuss how this values can be used anyway.

Number: 4      Author: Referee      Date: 05.12.2017 13:22:38

?? weird sentence, please simplify and rephrase.

Number: 5      Author: Referee      Date: 05.12.2017 13:22:38

which seasonal approach?

Number: 6      Author: Referee      Date: 05.12.2017 13:22:38

is named differently in plot, please homogenize.

[revised manuscript text omitted]

what does this mean? Is it explained?

| | Number: 2 | Author: Referee | | Date: 05.12.2017 13:28:13 |
|---|---|---|---|---|

is this a political and geographical correct phrase?

| | Number: 3 | Author: Referee | | Date: 05.12.2017 13:22:38 |
|---|---|---|---|---|

| | Number: 4 | Author: Referee | | Date: 05.12.2017 13:22:38 |
|---|---|---|---|---|

| | Number: 5 | Author: Referee | | Date: 05.12.2017 13:22:38 |
|---|---|---|---|---|

(Beijing, Shanghai, Guangzhou, Chongqing, Wuhan)

| | Number: 6 | Author: Referee | | Date: 05.12.2017 13:22:38 |
|---|---|---|---|---|

one word missing

| | Number: 7 | Author: Referee | | Date: 05.12.2017 13:22:38 |
|---|---|---|---|---|

what do you mean?

| | Number: 8 | Author: Referee | | Date: 05.12.2017 13:22:38 |
|---|---|---|---|---|

?

| | Number: 9 | Author: Referee | | Date: 05.12.2017 13:22:38 |
|---|---|---|---|---|

similar with respect to what?

| | Number: 10 | Author: Referee | | Date: 05.12.2017 13:22:38 |
|---|---|---|---|---|

I do not understand the argumentation

[revised manuscript text omitted]

**Number: 1**     Author: Referee     Date: 05.12.2017 13:25:25
please also indicate this in the figure caption and probably use the more common a.g.l. nomenclature

**Number: 2**     Author: Referee     Date: 05.12.2017 13:22:38
not SE ASIA.....

**Number: 3**     Author: Referee     Date: 05.12.2017 13:22:38
state some numbers.

**Number: 4**     Author: Referee     Date: 05.12.2017 13:22:38
,

**Number: 5**     Author: Referee     Date: 05.12.2017 13:22:38

**Number: 6**     Author: Referee     Date: 05.12.2017 13:28:29
one cannot see this in the plot. From ground-based observations, much higher dust top heights are observed. Can you explain the differences?

**Number: 7**     Author: Referee     Date: 05.12.2017 13:22:38
sorry, I do not understand. please rephrase!

**Number: 8**     Author: Referee     Date: 05.12.2017 13:22:38
,

**Number: 9**     Author: Referee     Date: 05.12.2017 13:22:38
I do not see that, please explain!

**Number: 10**     Author: Referee     Date: 05.12.2017 13:22:38
no sentence! What do you mean CoM and TH higher of Thar, or Thar more active than...?

**Number: 11**     Author: Referee     Date: 05.12.2017 13:22:38
higher

**Number: 12**     Author: Referee     Date: 05.12.2017 13:22:38
please change plot scale so that one can see that --> at least up to 6 km.

**Number: 13**     Author: Referee     Date: 05.12.2017 13:22:38

**Number: 14**     Author: Referee     Date: 05.12.2017 13:22:38
over

[revised manuscript text omitted]

Number: 1        Author: Referee        Date: 05.12.2017 13:22:38

Number: 2        Author: Referee        Date: 05.12.2017 13:22:38
in Table 1

Number: 3        Author: Referee        Date: 05.12.2017 13:22:38
?? I do not understand this...

Number: 4        Author: Referee        Date: 05.12.2017 13:22:38
you must indicate this within the plots. probably adding small maps and showing the band of interest!

Number: 5        Author: Referee        Date: 05.12.2017 13:22:38
maximum elevation of surface elevation? What is this?

Number: 6        Author: Referee        Date: 05.12.2017 13:22:38
why four?

Number: 7        Author: Referee        Date: 05.12.2017 13:22:38
what do you mean?

[revised manuscript text omitted]

**T** Number: 1   Author: Referee   Date: 05.12.2017 13:22:38
please rephrase scientific correctly.

**T** Number: 2   Author: Referee            Date: 05.12.2017 13:22:38
this is complete contradictory  to the dust top heights presented before.

**T** Number: 3   Author: Referee   Subject: Inserted Text   Date: 05.12.2017 13:22:38
,

**T** Number: 4   Author: Referee   Subject: Comment on Text   Date: 05.12.2017 13:22:38
this information was just given before.

**T** Number: 5   Author: Referee   Subject: Comment on Text   Date: 05.12.2017 13:22:38
I guess I know what this means but its not yet explained. But better to homogenize wordings in this paper. I.e. use only Calipso AOD.

[revised manuscript text omitted]

but also vertical distribution, right?
* * *
**Number: 2**     Author: Referee     Subject: Comment on Text     Date: 05.12.2017 13:22:38

I would explain this abbreviations in the conclusion again at least once.
* * *
**Number: 3**     Author: Referee     Subject: Comment on Text     Date: 05.12.2017 13:22:38

what do you mean with this, i.e. what is the magnitude of a feature?
* * *
**Number: 4**     Author: Referee     Subject: Comment on Text     Date: 05.12.2017 13:22:38

please state more precisely.
* * *
**Number: 5**     Author: Referee     Subject: Comment on Text     Date: 05.12.2017 13:22:38

please rephrase
* * *
**Number: 6**     Author: Referee     Subject: Comment on Text     Date: 05.12.2017 13:22:38

Now you should highlight your NEW results and presents some "cherries" concerning the vertical distribution. dust detected up to..... dust mainly below....
* * *
**Number: 7**     Author: Referee     Subject: Comment on Text     Date: 05.12.2017 13:22:38

I do not understand this sentence.
* * *
**Number: 8**     Author: Referee     Subject: Inserted Text     Date: 05.12.2017 13:22:38

,
* * *
[Figure]

[revised manuscript text omitted]

**Number: 1**     Author: Referee     Subject: Cross-Out   Date: 05.12.2017 13:22:38

**Number: 2**     Author: Referee     Subject: Inserted Text     Date: 05.12.2017 13:22:38

T

**Number: 3**     Author: Referee     Subject: Comment on Text     Date: 05.12.2017 13:22:38

you should clarify which product is used (conditional or climatological)

**Number: 4**     Author: Referee     Subject: Comment on Text     Date: 05.12.2017 13:22:38

You should plot these two figures below each other or behind each other and discuss clearly the differences and the resulting messages for atmospheric research

**Number: 5**     Author: Referee     Subject: Cross-Out   Date: 05.12.2017 13:22:38

[Figure]

[Figure]

Table 1: Domain statistics on mean Dust Optical Depth, max D_AOD / 95th percentile, dust Center of Mass (CoM) and Top Height (TH) (both in km a.s.e.) and number of dust profiles to the total number of cloud-free profiles, based on the period 01/2007-12/2015.

| | Mean D_AOD ± SD (Climatological) | D_AOD Max / Percentile 95% (Climatological) | Dust CoM ± SD (km / a.s.e.) (Conditional) | Dust Top Height ± SD (km / a.s.e.) (Conditional) | Nr of Dust in Nr of cloud-free |
|---|---|---|---|---|---|
| **Taklimakan / Gobi** | | | | | |
| DJF | 0.078 ± 0.135 | 1.802 / 0.327 | 2.31 ± 1.39 | 3.47 ± 2.01 | 0.74 |
| MAM | 0.193 ± 0.308 | 2.729 / 0.819 | 3.06 ± 1.43 | 1.01 ± 2.17 | 0.78 |
| JJA | 0.113 ± 0.232 | 2.504 / 0.529 | 3.19 ± 1.38 | 4.94 ± 1.8 | 0.68 |
| SON | 0.095 ± 0.18 | 2.488 / 0.401 | 2.58 ± 1.27 | 3.92 ± 1.77 | 0.73 |
| **Tibetan Plateau** | | | | | |
| DJF | 0.012 ± 0.037 | 0.758 / 0.062 | 6.01 ± 1.26 | 6.96 ± 1.42 | 0.31 |
| MAM | 0.028 ± 0.055 | 0.731 / 0.127 | 6.2 ± 1.21 | 2.76 ± 1.64 | 0.52 |
| JJA | 0.013 ± 0.033 | 0.676 / 0.068 | 5.99 ± 1.05 | 7.13 ± 1.3 | 0.39 |
| SON | 0.006 ± 0.023 | 0.631 / 0.032 | 6.08 ± 1.37 | 6.87 ± 1.45 | 0.24 |
| **SE China** | | | | | |
| DJF | 0.062 ± 0.104 | 1.769 / 0.254 | 1.73 ± 1.29 | 3.03 ± 1.84 | 0.79 |
| MAM | 0.108 ± 0.171 | 2.39 / 0.404 | 2.38 ± 1.47 | 4.35 ± 2.26 | 0.85 |
| JJA | 0.032 ± 0.064 | 1.06 / 0.145 | 1.9 ± 1.56 | 3.09 ± 2.13 | 0.69 |
| SON | 0.045 ± 0.081 | 1.329 / 0.19 | 1.6 ± 1.27 | 2.74 ± 1.71 | 0.72 |
| **Indian Peninsula** | | | | | |
| DJF | 0.043 ± 0.065 | 1.736 / 0.147 | 1.13 ± 0.9 | 2.13 ± 1.28 | 0.84 |
| MAM | 0.171 ± 0.188 | 1.944 / 0.521 | 1.79 ± 0.99 | 3.63 ± 1.44 | 0.93 |
| JJA | 0.199 ± 0.167 | 2.071 / 0.751 | 2.05 ± 1.22 | 3.72 ± 1.55 | 0.86 |
| SON | 0.075 ± 0.106 | 1.459 / 0.267 | 1.29 ± 0.87 | 2.54 ± 1.25 | 0.83 |
| **N Pacific** | | | | | |
| DJF | 0.026 ± 0.059 | 1.169 / 0.113 | 1.55 ± 1.54 | 2.49 ± 1.97 | 0.67 |
| MAM | 0.046 ± 0.085 | 1.596 / 0.196 | 2.29 ± 1.86 | 3.96 ± 2.7 | 0.79 |
| JJA | 0.007 ± 0.02 | 0.613 / 0.035 | 1.55 ± 1.95 | 2.29 ± 2.38 | 0.45 |
| SON | 0.012 ± 0.032 | 0.929 / 0.057 | 1.26 ± 1.49 | 2.06 ± 1.89 | 0.56 |
| **Indochina / Indonesia** | | | | | |
| DJF | 0.005 ± 0.016 | 0.562 / 0.022 | 0.84 ± 0.99 | 1.28 ± 1.11 | 0.39 |
| MAM | 0.005 ± 0.012 | 0.269 / 0.024 | 0.98 ± 1.16 | 1.47 ± 1.29 | 0.47 |
| JJA | 0.003 ± 0.01 | 0.383 / 0.018 | 1.19 ± 1.96 | 1.6 ± 2.07 | 0.33 |
| SON | 0.003 ± 0.012 | 0.712 / 0.016 | 1.02 ± 1.57 | 1.45 ± 1.71 | 0.38 |

Number: 1     Author: Referee          Date: 05.12.2017 13:22:38
please indicate this in the plots by changing the scale

Number: 2     Author: Referee          Date: 05.12.2017 13:22:38
as above

[Figure]

[Figure]

**Fig.1**

[Figure]

[Figure]

[Figure]

Fig.2

[Figure]

[Figure]

5      **Fig. 3**

Number: 1    Author: Referee    Date: 05.12.2017 13:22:38

Does it make sense  for regions of almost no dust occurrence?

Number: 2    Author: Referee    Date: 05.12.2017 13:22:38

Please us a different altitude scaling! Dust top height is much higher than the CoM

Number: 3    Author: Referee    Date: 05.12.2017 13:22:38

 what happens to values above scale maximum? Not plotted?

[Figure]

[Figure]

Fig. 4

Number: 1     Author: Referee     Date: 05.12.2017 13:22:38
please indicate deserts in plot if possible

Number: 2     Author: Referee     Date: 05.12.2017 13:22:38
please write dust extinction coeff.

[Figure]

[Figure]

**Fig. 5**

[Figure]

[Figure]

[Figure]

**Fig. 6**

[Figure]

[Figure]

5    Fig. 7

[Figure]

[Figure]

[Figure]

**Fig. 8**

---

## Author Comment (AC1) · 13 Dec 2017

**Response to Anonymous Referee #1**

**Comments to the Author**

**General comments**

**In this paper, the spatial and temporal evolution of desert dust aerosols over South-East Asia has been systematically investigated based on CALIPSO since it can provide much information about aerosols. However, I'm interesting to see this paper published before revised as below suggestion.**

The authors would like to thank the reviewer for the interesting and at the same time substantial comments and suggestions. We tried, and did our best, to incorporate the most suitable proposed changes and corrections in the revised manuscript, aiming to the improvement of the presented paper.
Following, you will find our responses that are addressed to the Editorial board and the reviewers too.

**Specific comments**

**1. Page 1, Line 1: "Dust aerosols have a significant role on climate through the direct radiative effect of absorption and scattering of solar and thermal terrestrial radiation".**
**I think you should add the reference:**
- **Huang, J., Fu, Q., Su, J., Tang, Q., Minnis, P., Hu, Y., Yi, Y., and Zhao, Q.: Taklimakan dust aerosol radiative heating derived from CALIPSO observations using the Fu-Liou radiation model with CERES constraints, Atmos. Chem. Phys., 9, 4011-4021, https://doi.org/10.5194/acp-9-4011-2009, 2009.**
- **Chen, SiYu, JianPing Huang, JingXin Li, Rui Jia, NanXuan Jiang, LiTai Kang, XiaoJun Ma, and TingTing Xie. 2017. "Comparison of Dust Emissions, Transport, and Deposition between the Taklimakan Desert and Gobi Desert from 2007 to 2011." Science China-Earth Sciences 60 (7):1338–55. https://doi.org/10.1007/s11430-016-9051-0.**

We agree with the reviewer that the manuscript and the discussion would improve by including the recommended references. References are added in the manuscript.

**2. In the paragraph 1, the semi-effect of dust should be also added. The effect can be seen from the references:**
**Huang, Jianping, Bing Lin, Patrick Minnis, Tianhe Wang, Xin Wang, Yongxiang Hu, Yuhong Yi, and J. Kirk Ayers. 2006. "Satellite-Based Assessment of Possible Dust Aerosols Semi-Direct Effect on Cloud Water Path over East Asia." Geophysical Research Letters 33 (19):L19802. https://doi.org/10.1029/2006GL026561.**

We agree with the reviewer that the manuscript and the discussion would improve by including the recommended references. References are added in the manuscript.

**3. Line 10: "airborne mineral dust is considered a significant atmospheric aerosol contributor", should be corrected into "...considered as a..."**

Corrected.

**4. Page 2, Line 29-32: "although passive satellite sensors provide information on the column properties of aerosols with adequate spatial and temporal resolution, they are bound to certain limitations, the major limitation being the lack of information on the three-dimensional distribution (vertical profile) of aerosols in the atmosphere, an important information for the assessment of the aerosols radiative forcing on climate as well as their contribution as IN and CCN (IPCC 2013)." This sentence is too long to understand it means, please rewrite it.**

The long sentence was re-written, in order to be easier to understand: "Although passive satellite sensors provide information on the column properties of aerosols with adequate spatial and temporal resolution, they are bound to certain limitations. The major limitation is the lack of aerosol information on the three-dimensional distribution (vertical profile), which consist of an important parameter for the assessment of the aerosols radiative forcing on climate as well as their contribution as IN and CCN (IPCC 2013).".

**5. Page 4, line 10: I think you should delete this words: "in order to discriminate the detected atmospheric features types into subtypes", because we have known the goal of the classification algorithm before this sentence.**

The authors are of the opinion that this section consists a methodology bridge to the pure-dust product, between the CALIPSO algorithm and aerosol subtype classification and the decoupling of the pure-dust component from the classified as dust and polluted dust aerosol layers by CALIPSO . Towards this goal and since the methodology section largely is based on the classification algorithm, the authors considered that these lines should not be deleted, but under consideration of the recommendation of the reviewer they are modified to:
"The Level-2 (L2) product consists the high-level quality products. More specifically, CALIPSO L2 algorithm classifies the detected layers into characteristic classes (Vaughan et al., 2009), namely into clear air, cloud, aerosol, stratospheric, surface, subsurface, totally attenuated or invalid feature types. The classification algorithm (Omar et al., 2009) utilizes the depolarization ratio and the magnitude of the attenuated backscatter signal, the height of the aerosol layers and the characteristics of the Earth's surface along the CALIPSO footprint (desert, ocean, snow/ice) in order to discriminate the detected atmospheric features types into subtypes".

**6. Page 4, line 14: the "2ox5o grid resolution" should be corrected into "2°x5°".**

Corrected.

**7. Page 4, line 17: "1ox1o" need to be corrected. The whole paper should be checked again.**

Corrected. The whole paper was checked again.

**8. Page 6, line 18: since you have said that the daytime minimum and nighttime minimum, what does the "minimum detectable AOD of 0.005" mean?**

According to the reviewer's recommendation the text is corrected to: "Regarding the uncertainties of the products, CALIOP L2 V3 is characterized by daytime minimum detectable backscatter of 0.0017±0.0003 km$^{-1}$sr$^{-1}$, nighttime minimum detectable backscatter of 0.0008±0.0001 km$^{-1}$sr$^{-1}$ and AOD of 0.005 (based on the minimum CALIOP 532 nm channel detection sensitivity, Winker et al., 2009)".

**9. Page 4, line 24: what does the "SAMUM" mean? Please write the full name.**

The text is modified according to the reviewer's recommendation: "During the SAharan Mineral dUst experiMent (SAMUM) 1 and 2 campaigns Saharan dust ...".

**10. Page 4, line 51: in this paragraph you introduce the methods of distinguish pure dust and non-dust. However, I still don't know the differences of the CALIPSO product of dust and polluted dust with the pure dust and non-dust. Since we can directly derive the dust extinction coefficient and profiles from the product, why don't you use it? And what about merits of the method to select the pure dust? What's the differences of the pure dust and dust products directly from CALIPSO L2?**

The CALIPSO V3 aerosol classification algorithm classifies the detected aerosol features as marine, dust, clean continental, polluted continental, polluted dust and smoke (Omar et al., 2009). Typical dust particle depolarization ratio values measured with lidars in field campaigns around the globe show values between 0.27 and 0.35 at 532 nm. Furthermore, the measurements show little variation independently of the source region, (e.g., Ansmann et al., 2011; Sakai et al., 2000; Liu et al., 2008b; Freudenthaler et al., 2009; Groß et al., 2011; Burton et al., 2013; Groß et al., 2013; Groß et al., 2015; Illingworth et al., 2015). Based on the dust depolarization ratio, a methodology has been established to discriminate the pure-dust component from mixtures of dust and non-dust aerosol layers (Tesche at al. 2009). In this methodology both the CALIPSO dust and polluted dust aerosol types are treated as mixtures of dust aerosols and non-dust aerosols. The methodology is applied and the final CALIPSO pure-dust product (the pure-dust component of the dust/polluted dust layers of CALIPSO) (Amiridis et al., 2013) are available to perform CALIPSO climatological studies (Marinou et al., 2017) and to develop interesting dust-related products (LIVAS-Amiridis et al., 2015).

**11. Page 5, line 30: please check this sentence of "The seasonal zonal distribution of the climatological and conditional dust extinction coefficient (Mm$^{-1}$)". If it's right to explain it.**

The reviewer is right, this was an editing error by the authors. The author's intension was to implement typographical symbol in order to introduce a list of CALIPSO products that would be used in the study and accordingly extensively discussed. Omitting the typographical symbol resulted in much confusion and we apologize for this mistake. The symbols have been restored, the list is clarified along with the sentence.

**12. Page 5, line30: I want to know whether the climatological dust extinction coefficient means the aerosol extinction without dust extinction coefficient since you write this sentence "This is accomplished by setting the dust extinction coefficient value of 0 km-1, for observations with non-dust aerosols". And the conditional dust product only has the dust extinction coefficient.**

The climatological extinction coefficient is computed by setting the extinction coefficient value of the non-dust aerosols to $0 \text{ km}^{-1}$, when averaging the profiles over a grid. The authors agree with the reviewer that this part of the manuscript was not clear, therefore it is re-written as follows:
"The climatological dust product is a measure of the average dust load over a geographical domain and is computed acknowledging only the contribution of the dust component in the atmosphere. Technically, this is accomplished by setting the extinction coefficient value of the non-dust aerosols to $0 \text{ km}^{-1}$, when averaging the profiles over a grid. The dust climatological product can be used for studies related to the contribution of dust to the total aerosol load over a period of time. In addition, the climatological dust product can be used in the evaluation of models related to dust transport and to radiative transfer models, in studies of dust-related physical processes (dust transport dynamics, CCN, IN), to investigate the effect of dust aerosols on ecosystems (dust deposition into the oceans) and to determine the dust aerosol load over highly industrialized and densely populated regions.
The conditional dust product is a measure of the average intensity of dust load over a geographical domain and is based explicitly on the dust profiles, hence ignoring completely non-dust aerosols. Technically, this is accomplished by setting the extinction coefficient value of the non-dust aerosols to not-a-number (NaN), when averaging the profiles over a grid. The conditional dust product is related to the intensity of the dust events."

**13. Page 7, line 33: what does the "N. China" mean?**

Corrected to: "Over N. China, for latitudes northern than 35° N, a similar pattern with respect to the features of dust contribution to the total aerosol load due to the dust aerosol emitted from the Taklimakan and Gobi deserts are observed".

**14. Page 8, line 11: from the figure 3, the differences of dust frequency in the four seasons are not clear, and the minimum in Fig. 3a is not obvious.**

Both the scale and the colormap of the dust frequency, CoM and TH are modified, according to the suggestion by the reviewer. Please see the figures below, before and after the adaptation of the figures.

**Before**

[Figure]

Fig. 3: Spatial distribution of dust occurrence [%], climatological pure-dust CoM (Center of Mass) and dust TH (Top Height) in km a.g.l., for each season over the domain between 65°-155° E and 5°-55° N and for the period 01/2007-12/2015.

**After**

[Figure]

[Figure]

**Fig. 3: Spatial distribution of dust occurrence [%], climatological pure-dust CoM (Center of Mass) and dust TH (Top Height) in km a.g.l., for each season over the domain between 65°-155° E and 5°-55° N and for the period 01/2007-12/2015.**

**15. Page 8, line 39: please explain the pattern of the dust transport since you said "however, the pattern reverses (Fig. 3i)"**

Corrected to: "During MAM, dust particles emitted from the Taklimakan and Gobi deserts are transported over C. China and the Pacific Ocean, while at the same time significant long-range transport of dust aerosols emitted from Thar Desert is not-observed (Fig. 3f). During JJA, however, the pattern reverses, with longer range of dust particles transported from Thar Desert over the Indian Peninsula, the Arabian Sea and the Bay of Bengal, while no significant dust transport of dust aerosol emitted from Taklimakan Desert is observed (Fig. 3i)."

---

## Author Comment (AC2) · 13 Dec 2017

**Response to Anonymous Referee #2**

**The paper by Proestakis et al. presents a 9-year climatology derived with mainly CALIOP measurements of the aerosol (dust) conditions over East and South Asia. The main focus is set on dust distribution but also non-dust aerosol is discussed. These novel results are thus of interest for atmospheric research and give a good overview of the dust distribution in this part of the world. Therefore in principle the paper is of interest for publication in ACP, however I recommend major revisions before it can be published. This is further explained below.**

The authors would like to thank the referee for the interesting and at the same time substantial constructive comments and suggestions. We tried, and did our best, to incorporate the proposed changes and corrections in the revised manuscript, aiming to the improvement of the presented paper. Following, you will find our responses that are addressed to the Editorial board and the reviewers as well.

**Major comments**

**In my opinion the naming of the study area is misleading. Even though South-East Asia may not be a protected phrase, many people have a different understanding concerning the region called this way. See for example:**
**https://en.wikipedia.org/wiki/Southeast_Asia**
**Therefore, I recommend to find a better name for the study area (e.g. South and East Asia or whatever) to avoid confusion and change the title and text accordingly.**

According to the reviewer's recommendation the name of the study area is changed from "Southeast Asia" to "South and East Asia", both in the title and the text.

**The difference between the climatological and conditional dust product needs be more discussed in both, the methodology section (I just understood the difference when reading Marinou 2017 but as this is essential it should be explained more explicitly here), but also in the result section. The reader is left alone with contradictory statements, like for example the dust top height which seems to be completely different between the two products. Therefore it should be clearly discussed:**
**(1) Which product can be used for which purpose?**
**(2) Does it make sense to use this two different products, if yes why and for what?**
**(3) What we can learn from the two products presented here with respect to South and East Asia.**

Both the "Data and Methodology" and the "Results" Sections have been revised and re-written according to the recommendation of the reviewer. To be more specific, (1) and (2) have been re-written and extended as follows:

The seasonal zonal distribution of the climatological and conditional dust extinction coefficient ($Mm^{-1}$).

The climatological dust product is a measure of the average dust load over a geographical domain and is computed acknowledging only the contribution of the dust component in the atmosphere. Technically, this is accomplished by setting the extinction coefficient value of the non-dust aerosols to 0 km-1, when averaging the profiles over a grid. The dust climatological product can be used for studies related to the contribution of dust to the total aerosol load over a period of time. In addition, the climatological dust product can be used in the evaluation of models related to dust transport and to radiative transfer models, in studies of dust-related physical processes (dust transport dynamics, CCN, IN), to investigate the effect of dust aerosols on ecosystems (dust deposition into the oceans) and to determine the dust aerosol load over highly industrialized and densely populated regions.

The conditional dust product is a measure of the average intensity of dust load over a geographical domain and is based explicitly on the dust profiles, hence ignoring completely non-dust aerosols. Technically, this is accomplished by setting the extinction coefficient value of the non-dust aerosols to not-a-number (NaN), when averaging the profiles over a grid. The conditional dust product is related to the intensity of the dust events.

In addition to the above a Flowchart is provided according to the reviewer's suggestion in the end of the ""Data and Methodology" Section (comment 4.3).

(3) is additional included. In general the findings are summarized in "Summary and conclusions" Section, which is re-written.

**Conclusion: The current conclusion is not very informative. Thus, it should be really overworked to highlight new things and discuss what lessons have been learned, i.e. what are new results or newly gathered knowledge or does your study just confirm former studies etc...**

The authors agree with the reviewer, the conclusion section was re-written in order to provide more information and highlights of the study. Below we provide a part of the conclusion which was vastly rephrased and extended:

"In this work, CALIPSO is used to provide a multiyear 4-D climatology of desert dust aerosols over South and East Asia at a spatial resolution of 1ox1o deg grids. An optimized dust aerosol product, developed using CALIOP backscatter and particle depolarization ratio, along with a regional correction on dust lidar ratio suitable for Asian dust is used. The optimized product is utilized to provide the horizontal and vertical distribution along with the temporal evolution of dust aerosols over a 9-year period (01/2007-12/2015).

Regarding the horizontal distribution of Aerosol Optical Depth (AOD), Dust Aerosol Optical Depth (D_AOD) and Non-Dust Aerosol Optical Depth (Non-Dust AOD), our analysis shows similar patterns between all four seasons, although the magnitude of the observed features varies with season. High values of Non-Dust AOD are consistently observed over the heavily industrialized and densely populated regions of China and India (Non-Dust AOD > 0.5). In addition to the anthropogenic densely populated areas of South and East Asia, the major sources of dust aerosols, namely the Taklimakan, Gobi and Thar Deserts are clearly mapped through the systematic high D_AOD values throughout the year. The magnitude though of the D_AOD observed features is subject to high seasonality, ranging between D_AOD 0.2 during winter and higher than 0.6 during spring and summer seasons. Maximum activity of Gobi and Taklimakan deserts is observed during spring, while the highest activity of Thar Desert is

during summer. The seasonality of the dust transport pathways is additionally well-captured. Dust transport over the Indian Peninsula is more pronounced during spring and summer, while over China similar patterns of a persistent dust aerosol background is evident throughout the year, with a peak during spring when the dust transport across the Pacific Ocean is at its maximum.

Regarding the vertical distribution of dust aerosols, the Center of Mass (CoM), Top Height (TH) and the mean Dust Extinction Coefficient profiles (Climatological and Conditional) are implemented to provide, together with the horizontal distribution, the full three-dimensional structure of dust aerosols and the atmospheric dust transport pathways over the entire South and East Asia. Based on the synergy of CoM, TH and the CALIPSO dust extinction profiles two distinct dust transport pathways over South and East Asia are observed: a the Trans-Pacific belt between 25o and 45o N and a second one, extending from Thar Desert towards the Bay of Bengal and the Arabian Sea. Both zones of dust transport are subject to high seasonality. Highest dust aerosol transport from the Taklimakan Desert towards the Pacific Ocean is observed during spring, while dust aerosol transport from the desert of Thar and across the Indian Subcontinent is more pronounced during summer.

Regarding the temporal evolution of AOD and D_AOD between 01/2007 and 12/2015, the analysis showed statistically significant positive short-term AOD trends over the Indian Peninsula (0.01 yr-1), NW China (0.007 yr-1) and E China (0.01 yr-1), whereas our study shows negative short-term AOD trends over SE China (-0.007 yr-1). CALIPSO positive AOD trends are found over the broader central and eastern Indian Peninsula (0.01 yr-1). The CALIOP observed trends between 01/2007 and 12/2015 are generally in qualitative agreement with the derived MODIS AOD trends over large domains of South and East Asia, although the short-term trends disagree over specific regions. The CALIOP and MODIS trends though are interpreted and compared with caution, since the samples of the datasets are non-uniform. ”

**Please check spelling and grammar intensively again. There are many sentences which are fractals, i.e. words are missing. Furthermore, many commas are missing, considering the bunch of co-workers this should be no problem.**

The authors have gone through the entire manuscript again to check for spelling and grammar again. At this point the authors would like to thank the reviewer once more for the substantial contribution towards the direction of improving the overall manuscript.

**Minor comments**

**Page 1.1:** According to the reviewer's recommendation the name of the study area is changed from "Southeast Asia" to "South and East Asia", both in the title and the text.

**Page 1.2:** According to the reviewer's suggestion, two sentences are used to simplify the initial sentence: "To distinguish desert dust from total aerosol load we apply a methodology developed in the framework of EARLINET (European Aerosol Research Lidar Network). The methods involves the use of particle linear depolarization ratio and updated lidar ratio values suitable for Asian dust, from multiyear CALIPSO observations (01/2007-12/2015)."

**Page 1.3:** Wavelength is included: "532 nm".

**Page 1.4:** Suggestion is included.

**Page 2.1:** Suggestion is included, the text is modified accordingly.
**Page 2.2:** The text is corrected: from "Major dust Asian" to "Major Asian dust".
**Page 2.3:** According to the suggestion the text is modified: from "CALIOP measures total attenuated backscatter signals at …" to "CALIOP measures total attenuated backscatter at …"

**Page 3.1:** According to the suggestion the text is rephrased: "Using this classification they either did not take into consideration the dust component of the classified as, polluted dust aerosol subtype, or they defined as "dust" both the dust and polluted dust aerosol subtypes (hence including the non-dust component of polluted dust)".
**Page 3.2:** Suggestion is included.
**Page 3.3:** Suggestion is included: "… this new pure dust …".
**Page 3.4:** Suggestion is included: "… pure dust product …".
**Page 3.5:** The text is corrected: from "laser" to "lidar".
**Page 3.6:** Since according to the reviewer the sentence was not clear the text is rephrased to: "CALIOP transmits linear polarized light, while a telescope of 1 m diameter collects the backscatter component backscattered by the atmosphere.".

**Page 4.1:** The text is modified according to the reviewer's recommendation and the different feature type classes are included: "The Level-2 (L2) product consists the high-level quality products. More specifically, CALIPSO L2 algorithm classifies the detected layers into characteristic classes (Vaughan et al., 2009), namely into clear air, cloud, aerosol, stratospheric, surface, subsurface, totally attenuated or invalid feature types."
**Page 4.2:** The manuscript is cross-checked and the aerosol subtype classification scheme is not explained before this part in the data and methodology section. Part of the paragraph though is rephrased: "The classification algorithm (Omar et al., 2009) utilizes the depolarization ratio and the magnitude of the attenuated backscatter signal, the height of the aerosol layers and the characteristics of the Earth's surface along the CALIPSO footprint (desert, ocean, snow/ice) in order to discriminate the detected atmospheric features types into subtypes. The atmospheric features types classified as aerosols are further distinguished into specific aerosol subtypes (Clean Marine, Dust, Clean Continental, Polluted continental, Polluted Dust and Smoke)."
**Page 4.3:** According to the suggestion the following Flowchart, diagrammatic representation from the CALIPSO data to the Pure-Dust product and the Climatological/Conditional Dust products used in the study is added to the end of the "Data and Methodology" Section.

[Figure]

Fig. 2: Flowchart of the CALIPSO Pure-Dust, Conditional Dust Extinction Coefficient and Climatological Dust Extinction Coefficient products.

**Page 4.4:** The text is corrected: from "… with cloud observations are filtered from …" to "… with cloud observations are filtered out from …".

**Page 4.5:** According to the suggestion the text is modified: from "… value of the pure dust component, $\beta_\perp$ …" to "… value of the pure dust component in the aerosol mixture, $\beta_\perp$ …"

**Page 4.6:** According to the recommendation by the reviewer the following two references are added: "Omar et al., 2009".

**Page 5.1:** The authors agree with the reviewer that the paragraph in the beginning of the "Data and Methodology" section is confusing, therefore the entire paragraph was moved to the end of the "Introduction Section" at the part of the description of the study domain.

**Page 5.2:** According to the reviewer's recommendation the name of the study area is changed from "Southeast Asia" to "South and East Asia".

**Page 5.3:** The backscatter coefficient.

**Page 5.4:** The text is corrected: from "extinction coefficient" to "backscatter coefficient".

**Page 5.5:** The reviewer is right, this was an editing error. The author's intension was to implement typographical symbol in order to introduce a list of CALIPSO products that would be used in the study and accordingly discussed. Omitting the typographical symbol resulted in much confusion to both the reviewers and we apologize for this mistake. The symbols have been restored and the list is clarified along with the sentence.

**Page 5.6:** Done (major comment and comment 4.3).

**Page 5.7:** Done (major comment and comment 4.3).

**Page 5.8:** According to the reviewer's recommendation the sentence is modified to include which product: "Validation of the pure dust aerosol product against …".

**Page 5.9:** According to the reviewer's recommendation the sentence is modified to include the domain: "… observations over northern Africa and Europe show …".

**Page 5.10:** The sentence did not make sense since it is part of a list which was omitted. Corrected through the introduction of the list and the typographical symbols (comment 5.5).

**Page 6.1:** Full stop added.

**Page 6.2:** Wavelength is included: "532 nm".

**Page 6.3:** Discuss AOD uncertainty is approximately 100% close to the surface (Marinou et al., 2017).

**Page 6.4:** According to the suggestion from the reviewer the sentence "Additional, uncertainty which propagates into the D_AOD product is introduced due to the depolarization ratio of the non-dust aerosols, coupled into the polluted-dust aerosol subtype." is rephrased to: "In addition, as it is already mentioned, both aerosol types classifies by CALIPSO as dust or polluted dust are a mixture of a dust component and a non-dust component. Thus another source of uncertainty in the decoupling of the dust component from the total aerosol load is the lack of information regarding the non-dust component in the aerosol mixture, due to the low depolarization ratio values of the non-dust aerosol subtypes (Omar et al., 2009)."

**Page 6.5:** According to the suggestion from the reviewer the sentence "... the seasonal approach is selected ..." is rephrased to: "… in this section we present and discuss the horizontal distribution of aerosols and dust over South and East Asia per season."

**Page 6.6:** According to the suggestion from the reviewer the entire paper is homogenised. "ΔOD" is replaced to "Non-Dust AOD".

**Page 7.1:** Satellite-based remote sensing, both passive and active, is highly sensitive to the presence of clouds. Regarding CALIOP, the nadir-viewing lidar measurements and orbital characteristics of CALIPSO result in a low frequency of overpasses over each region and consequently the significant fewer observations with respect to passive sensors. Therefore, in order to provide meaningful climatologies on a regional scale, long-term and multiyear CALIOP observations are required. Even though over specific regions which are characterized of extensive cloud coverage of dense cloud (Bay of Bengal, Indonesia, N. Pacific Ocean), the number of observations in the sample is even sparser, due to attenuation in dense clouds. Therefore rare regional events are not well captured over regions of extensive cloud coverage, resulting in weighted values toward regions of less extensive cloud coverage (Winker et al., 2013).

**Page 7.2:** Indochina is a political and geographical correct term. Indochina, originally Indo-China, is a geographical term originating in the early nineteenth century and referring to the continental portion of the region now known as Southeast Asia. The name refers to the lands historically within the cultural influence of India and China, and physically bound by the Indian Subcontinent in the west and China in the north. It corresponds to the present-day areas of Myanmar, Thailand, Laos, Cambodia, Vietnam, and (variably) peninsular Malaysia and Singapore. See for example: https://en.wikipedia.org/wiki/Indochina

**Page 7.3:** According to the suggestion from the reviewer the sentence is modified to: "Over China similar geographical patterns in the horizontal distribution of aerosols are evident between all four seasons, with larger Non-Dust AOD values over the major sources of anthropogenic aerosols (Beijing, Shanghai, Guangzhou, Chongqing, Wuhan) such as urban clusters (Beijing, Shanghai, Guangzhou, Chongqing, Wuhan) (Kourtidis et al., 2015) and high D_AOD values over the deserts of Taklimakan and Gobi (Che et al. 2014, 2015)".

**Page 7.4:** Done (comment 7.3).

**Page 7.5:** Done (comment 7.3).

**Page 7.6:** Corrected: "… high D_AOD values …".

**Page 7.7:** For example to the north of the plateau of Tibet, during the period between March and May the strong surface winds which develop over the Mongolian Plateau create favourable mechanisms of extreme dust events (Bory et al., 2003; Yu et al., 2008). This feature is evident throughout the year, although more pronounced during spring. By activation of the deserts the authors mean the creating of favourable conditions for dust generation and injection in higher altitude in the atmosphere.

**Page 7.8:** The sentence is rephrased to: "Over China, for latitudes northern than 35o N, a similar pattern with respect to the features of dust contribution to the total aerosol load due to the dust aerosol emitted from the Taklimakan and Gobi deserts are observed. More specific, a persistent dust aerosol background is evident during all seasons, with a peak during MAM (Fig. 3f)."

**Page 7.9:** Rephrased (comment 7.8).

**Page 7.10:** This is in line with previous studies, reporting rare events of dust transport over Himalayas (Huang et al., 2007; Liu et al., 2008b; Yumimoto et al., 2009). The region to the North of Taklimakan, Gobi and Mongolia is also characterized by low values of AOD and D_AOD, except during MAM (Fig. 3f, h). The high dust aerosol load observed to the east of the major dust aerosol source of Taklimakan (D_AOD values greater than 0.3) and the high percentage of D_AOD with respect to the total AOD indicate a strong eastward transport of both dust (Fig. 3f) and anthropogenic aerosols (Fig. 3g). The paragraph is modified accordingly.

**Page 8.1:** Both suggestions are implements, the a.g.l. is added to the figure caption and the common nomenclature are used.

**Page 8.2:** According to the reviewer's recommendation the name of the study area is changed from "Southeast Asia" to "South and East Asia".

**Page 8.3:** According to the recommendation by the reviewer we have included some numbers. More precisely the following part is added: "Lower frequencies of dust occurrence, which still exceed 70%, are also evident over east China and south-eastern

India. Conversely over Indochina and Indonesia the occurrence of dust is particularly low, especially during summer (Fig. 4g) and autumn (Fig. 4j). To be more specific, values of dust occurrence percentage between 50% and 60% over Thailand and Cambodia, 40% to 60% over Laos and Vietnam, ~60% over SE China, and lower than 40% over Malaysia and Philippines are observed, during JJA and SON.".

**Page 8.4:** Comma added.

**Page 8.5:** "s" deleted.

**Page 8.6:** The authors are not sure in which paper the reviewer is referring to, though we suspect that the differences most probably are related to the definition of the TH and CoM, probably to differences in the above ground level or above sea level reported profiles, maybe to different seasonality, sensor detection limits, differences in the techniques applied, to different samples. There are just many factors that may result in the different observations.

**Page 8.7:** The sentence is rephrased according to the suggestion of the reviewer, since the meaning was not clear. The sentence is rephrased from: "In addition, the observed gradient in the horizontal distribution of D_AOD values between the sources and the Pacific Ocean (Fig. 3f), parallel to the ubiquitous dust layer and the high dust TH (Fig. 4f), are an indicator of the longer range of transport of lower concentration of dust particles." to "In addition, a decreasing west-to-east D_AOD gradient is observed over N. China, between the dust sources over Taklimakan and Gobi and the Pacific Ocean (Fig. 3f). The decreasing gradient of TH is less pronounced during MAM, when dust aerosol are injected as high as 10 km height (a.s.l.) and transported longer distances over the Pacific Ocean (Fig. 4f).".

**Page 8.8:** Comma added.

**Page 8.9:** Since according to the reviewer the meaning was not clear, the paragraph was rephrased accordingly: "during MAM, dust particles emitted from the Taklimakan and Gobi deserts are transported over C. China and the Pacific Ocean, while at the same time significant long-range transport of dust aerosols emitted from Thar Desert is not-observed (Fig. 4f). During JJA, however, the pattern reverses, with longer range transport of dust particles from Thar Desert over the Indian Peninsula, the Arabian Sea and the Bay of Bengal, while no significant dust transport of dust aerosol emitted from Taklimakan Desert is observed (Fig. 4i).".

**Page 8.10:** Sentence corrected (comment 8.9).

**Page 8.11:** "larger" is replaced by "higher" according to the recommendation by the reviewer.

**Page 8.12:** Adapted (comment 27.1).

**Page 8.13:** Comma deleted.

**Page 8.14:** "into" is replaced by "over" according to the recommendation by the reviewer.

**Page 9.1:** "the" deleted.

**Page 9.2:** Sentence is modified accordingly: "… provided in Table 1".

**Page 9.3:** The pure dust component is of interest to regions where dust aerosols are present. In the domain of South and East Asia, such domains are East China and India. East China is affected from dust aerosol transport from the Taklimakan and Gobi, while India is affected from dust transport from Thar Desert and the Arabian Peninsula. The

significance of dust load is even larger due to the anthropogenic emissions of the densely populated provinces of China and the regions of India.

**Page 9.4:** Initially we had a column of the study domain, encompassing the different domains. In general, after initial submission, we have tried to move the individual figures as close to each other as possible, otherwise the individual plots won't exceed stamp size in the final paper and the reader won't be able to recognise a thing. Towards this need, we had to omit as much redundant information from the initial submission as possible, including the map plots which indicated the bands of interest, since the reader can take this information from Figure 1.

**Page 9.5:** Corrected: "The continuous and dashed lines correspond to the average elevation of the surface level and to the average maximum elevation respectively.".

**Page 9.6:** This threshold is arbitrarily selected to filter out low dust aerosol extreme cases scenarios in order to limit the influence of rare events on climatology. To be clearer, the following was added: of the pure dust climatological extinction coefficient, arbitrarily selected, in order to avoid presenting extreme rare events in high altitudes at the same time with climatological values close to the surface level

**Page 9.7:** Indeed the sentence "Regarding the Taklimakan Desert, this region is a very prolific arid area encompassed by Tarim Basin" is clarified to "Taklimakan Desert, consists a very arid area encompassed by Tarim Basin".

**Page 10.1:** Indeed, the reviewer is right, the sentence is modified accordingly: from "during MAM lofted of dust extinction coefficient ..." to "… during MAM dust aerosol layers are detected …".

**Page 10.2:** According to the opinion of the authors, the CoM should not be plotted together with the climatological dust extinction coefficient, and the reason is threefold. First of all, the purpose of the climatological dust product is to answer the question, which is the aerosol load over a specific region on a climatological basis, while the purpose of the CoM is to provide the climatological center of mass of the average dust profile. A rare example is related to a region where dust is never present. Over such a region the climatological dust profile is a well-defined profile of zeros. On the contrary, by mathematical definition, the CoM over such profile cannot be computed (NaN). The second reason is related to the representativeness. An example over the study domain of South and East Asia is the Himalayas orographic barrier. Extreme and very rare events of dust transport over Himalayas have been reported. The CoM plotted together with the climatological dust extinction coefficient profiles would create to the reader a misleading view of frequent dust transport over this region, due to the lack of the information of how many events are used to compute the profile and the CoM. The third reason is related to the fact that the CoM is always defined above ground level. Due to the complex surface orography of South and East Asia, the mean surface elevation of the used regions sometimes may be above the CoM. For example Tarim Basin, the basin that encompasses the Taklimakan Desert has an elevation approximately 2 km higher than the Taklimakan desert elevation. Thus during the non-active seasons the dust CoM would be below or very close to the mean Surface elevation of the region.

**Page 10.3:** Indeed, the sentence is modified accordingly from "... attributed the gravitational settling of aerosols and to dry and wet deposition (Colarco et al., 2003)."

to "... attributed to both dry and wet deposition processes that remove dust aerosol from the atmosphere (Colarco et al., 2003).".

**Page 10.4:** Indeed, the reviewer is right, the sentence is modified accordingly: from "MAM a lofted layer of dust climatological extinction coefficient up to 25 Mm$^{-1}$ is observed ..." to "…MAM a lofted layer of dust aerosols that yields climatological extinction coefficient up to 25 Mm$^{-1}$ is observed …".

**Page 10.5:** The Climatological Total Extinction Coefficient close to the surface over the densely populated regions of East China, especially during MAM when a large component of the AOD is related to dust aerosols transported from Taklimakan and Gobi deserts is as high as 200 Mm$^{-1}$. For more information: de Leeuw, G., Sogacheva, L., Rodriguez, E., Kourtidis, K., Georgoulias, A. K., Alexandri, G., Amiridis, V., Proestakis, E., Marinou, E., Xue, Y., and van der A, R.: Two decades of satellite observations of AOD over mainland China, Atmos. Chem. Phys. Discuss., https://doi.org/10.5194/acp-2017-838, in review, 2017.

**Page 11.1:** The difference between climatological and conditional is improved and adapted and already discussed.
**Page 11.2:** "coefficient" deleted.

**Page 12.1:** Since according to the reviewer it was not clear through the sentence where the features of dust transport pathways are observed in Figure 6, the sentence is modified accordingly: "To the north and east of the Tibetan Plateau two distinct eastward pathways of dust transport are evident: (1) a northern flow that propagates towards the Yellow Sea and the Pacific Ocean (Uno et al., 2009) (Fig. 6f, g) and (2) a southern flow that occurs over central China (Kuhlmann and Quaas, 2010) (Fig 6k, o)."
**Page 12.2:** Corrected. "Both transport pathways are observed at the middle and upper troposphere …".
**Page 12.3:** Indeed the authors agree with the reviewer that the paragraph was not written in a clear way to help the reader to navigate along the figures and easily absorb the information provided. Thus the entire paragraph was rewritten: "To the north and east of the Tibetan Plateau two distinct eastward pathways of dust transport are observed: (1) a northern flow that propagates towards the Yellow Sea and the Pacific Ocean (Uno et al., 2009) and (2) a southern flow that occurs over central China (Kuhlmann and Quaas, 2010). The northern flow is mostly evident during winter (Fig. 3d), while the southern transport pathway over C. China is more prominent during spring (Fig. 3h). Figure 6 provides information on the vertical distribution and depth of the two dust transport pathways. Both transport pathways are observed at the middle and upper troposphere, indicated by dust conditional coefficient values as high as 20 Mm$^{-1}$, observed at altitude up 10 km a.s.l. Another noticeable feature is that the vertical intensity of the transported dust aerosol plumes is subject to high spatial and seasonal variability. Decreasing values of both dust aerosol climatological and conditional values are observed with increasing distance from the dust sources of Taklimakan and Gobi deserts towards and over the Pacific Ocean.".
**Page 12.4:** Comment 12.3
**Page 12.5:** Comma deleted.
**Page 12.6:** Comment 12.3

**Page 12.7:** The reviewer is right. "(as discussed in Section Data and Methodology)" is added.

**Page 12.8:** According to the different particle depolarization values, a mean value of 0.03 value is used for the non-spherical aerosol types. This introduces uncertainties since the 0.03 is a mean value, ±0.01.

**Page 12.9:** "particle" added.

**Page 12.10:** Indeed the sentence is rephrased: "Figure 7 shows the vertical, horizontal and seasonal variability of the average particle depolarization ratio of the cases classified by CALIOP as dust or polluted dust aerosol subtypes based on nine years of CALIPSO observations (01/2007-12/2015) and for five zones of 10° latitudinal interval, between 5° and 55° N.".

**Page 12.11:** The misleading sentence is rephrased according to the reviewer's recommendation from "In general, to the north of Himalayas (Fig. 7e-h), a structure of three different height ranges is evident." to: "In general, to the north of Himalayas, low values of particle depolarization ratio are observed close to the surface, while particle depolarization ratio increases with increasing height (Fig. 7e-h).".

**Page 13.1:** Rephrased to: "Over Thar Desert the average particle depolarization ratio of cases classified as dust or polluted dust by the CALIPSO classification algorithm yield average depolarization values greater than 25% throughout the year".

**Page 13.2:** Height in Fig. 4 are defined above ground level while at Fig. 5, 6 and 7 characteristic are above sea level.

**Page 13.3:** Comma added.

**Page 13.4:** The sentence is deleted after cross-checking.

**Page 13.5:** The entire manuscript is homogenised accordingly.

**Page 14.1:** Over the manuscript frequently the authors have discussed the horizontal and vertical distribution of AOD and dust aerosol respectively. Both MODIS Aqua and CALIPSO CALIOP are utilized. Both sensors utilize different algorithms for cloud screening for cloud fraction and extensive cloudiness prevent retrievals over specific domains (Bay of Bengal and monsoon period). These figures, although not extensively discusses, are in the manuscript for completeness reason, for a reviewer to have a full overview of the datasets utilized.

**Page 14.2:** According to the reviewer's recommendation the sentence is rephrased from "Strong statistical increase is observed by MODIS over the Arabian Sea (0.01 yr$^{-1}$), while not statistical significant positive AOD550nm trends are present over the Bay of Bengal (0.002 yr$^{-1}$)." to "MODIS shows not statistical significant AOD increasing trends of the order of (0.002 yr$^{-1}$) over the Bay of Bengal and strong positive statistical significant trends over the Bay of Arabian Sea (0.01 yr$^{-1}$). The strongly increasing AOD trend over the Arabian Sea though is not corroborated by CALIOP observations.".

**Page 14.3:** The abbreviation is removed according to the reviewer's suggestion.

**Page 15.1:** Indeed, the reviewer is right. Based on CALIOP both the horizontal and vertical distribution of aerosol can be studied.

**Page 15.2:** According to the reviewer's recommendation the abbreviations are explained one last time in the beginning of the "Summary and Conclusions" section. To be more specific the following phrase is modified: "Our analysis shows similar

patterns in the horizontal distribution of Aerosol Optical Depth (AOD), Dust Aerosol Optical Depth (D_AOD) and Non-Dust Aerosol Optical Depth (Non-Dust AOD) between all four seasons ...".

**Page 15.3:** The authors mean, the range of the observations, how high/low the observed values are.

**Page 15.4:** Corrected.

**Page 15.5:** According to the reviewer's recommendation the sentence is rephrased: "The CoM, TH and the mean extinction coefficient profiles are implemented in order to provide the vertical dust aerosol distribution, and together with the horizontal distribution of AOD and D_AOD to provide in the end the three-dimensional structure of dust aerosol over South and East Asia and the atmospheric dust aerosol transport pathways.".

**Page 15.6:** "The Summary and Conclusions" Section is adapted according to the reviewer's suggestion (major comment).

**Page 15.7:** The sentence is rephrased. Actually the entire "Summary and Conclusions" Section is modified. The Angstrom Exponent and the spectral dependence of AOD is crucial since it can be used as a fingertip for identifying coarse aerosol types and different aerosol cases (e.g. dust or non-dust cases)

**Page 15.8:** Comma added.

**Page 26.1:** "Under" is deleted.

**Page 26.2:** Capital "T" is used according to the reviewer's correction.

**Page 26.3:** Clarified. "Climatological" is added.

**Page 26.4:** Indeed the two figures it is better to be plotted together, the authors totally agree with the reviewer's recommendation.

**Page 26.5:** Corrected. Word "Geographical" is deleted from the caption of figures 5, 6 and 7.

**Page 27.1:** Both the scale and the colormap are modified, according to the suggestion by the reviewer. Please see the figures below, before and after the adaptation of the figures.

**Page 27.2:** Both the scale and the colormap are modified, according to the suggestion by the reviewer. Please see the figures below, before and after the adaptation of the figures. In addition the phrase "please note the different height scale" is added to the manuscript.

**Before**

[Figure]

[Figure]

**Fig. 4: Spatial distribution of dust occurrence [%], climatological pure-dust CoM (Center of Mass) and dust TH (Top Height) in km a.g.l., for each season over the domain between 65°-155° E and 5°-55° N and for the period 01/2007-12/2015.**

**After**

[Figure]

| Dust Cases / Overpasses | Dust Center of Mass | Dust Top Height |
| --- | --- | --- |

**Fig. 4: Spatial distribution of dust occurrence [%], climatological pure-dust CoM (Center of Mass) and dust TH (Top Height) in km a.g.l., for each season over the domain between 65°-155° E and 5°-55° N and for the period 01/2007-12/2015.**

**Page 30.1:** Since CALIPSO CALIOP classifies layers as dust or polluted dust over these domains, even if the frequency is low - as indicated by the question of the reviewer, it makes sense to decouple those few cases to the pure-dust and non-dust components of the detected aerosol mixtures. The need is supported also by the evidence that dust sources are not only natural but anthropogenic activity may also lead to dust particles in the atmosphere, over domains where no natural dust is supposed to be observed through intercontinental transport. Chen, S., Huang, J., Jiang, N., Zang, Z., Guan, X., Ma, X., Jia, Z., Zhang, X., Zhang, Y., Huang, K., Xu, X., Zhang, G., Li, J., Yang, R., and Liao, S.: Estimations of anthropogenic dust emissions at global scale from 2007 to 2010, Atmos. Chem. Phys. Discuss., https://doi.org/10.5194/acp-2017-890, in review, 2017.
**Page 30.2:** Suggestion is implemented – comment 27.1.
**Page 30.3:** The scale of figures are adapted according to the recommendation of the reviewer in order to include both higher values (comments 27.1, 27.2, 30.3)

**Page 31.1:** The authors are of the opinion that any redundant information added on the figures 5, 6 and 7 would make it harder to read features on them. Thus the authors have not included marks indicating the desert regions on the figures, since a reader can take the information of the regions of interest and the locations of the deserts from Figure 1.
**Page 32.2:** According to the reviewer's recommendation, the caption of the colormap is changed to: "Dust Extinction Coefficient".

---

## Referee Report (RR1)

**Response to Anonymous Referee #1**

**Comments to the Author**

**General comments**

**In this paper, the spatial and temporal evolution of desert dust aerosols over South-East Asia has been systematically investigated based on CALIPSO since it can provide much information about aerosols. However, I'm interesting to see this paper published before revised as below suggestion.**

The authors would like to thank the reviewer for the interesting and at the same time substantial comments and suggestions. We tried, and did our best, to incorporate the most suitable proposed changes and corrections in the revised manuscript, aiming to the improvement of the presented paper. Following, you will find our responses that are addressed to the Editorial board and the reviewers too.

**Specific comments**

**1. Page 1, Line 1: "Dust aerosols have a significant role on climate through the direct radiative effect of absorption and scattering of solar and thermal terrestrial radiation".**
**I think you should add the reference:**
- **Huang, J., Fu, Q., Su, J., Tang, Q., Minnis, P., Hu, Y., Yi, Y., and Zhao, Q.: Taklimakan dust aerosol radiative heating derived from CALIPSO observations using the Fu-Liou radiation model with CERES constraints, Atmos. Chem. Phys., 9, 4011-4021, https://doi.org/10.5194/acp-9-4011-2009, 2009.**
- **Chen, SiYu, JianPing Huang, JingXin Li, Rui Jia, NanXuan Jiang, LiTai Kang, XiaoJun Ma, and TingTing Xie. 2017. "Comparison of Dust Emissions, Transport, and Deposition between the Taklimakan Desert and Gobi Desert from 2007 to 2011." Science China-Earth Sciences 60 (7):1338–55. https://doi.org/10.1007/s11430-016-9051-0.**

We agree with the reviewer that the manuscript and the discussion would improve by including the recommended references. References are added in the manuscript.

**2. In the paragraph 1, the semi-effect of dust should be also added. The effect can be seen from the references:**
**Huang, Jianping, Bing Lin, Patrick Minnis, Tianhe Wang, Xin Wang, Yongxiang Hu, Yuhong Yi, and J. Kirk Ayers. 2006. "Satellite-Based Assessment of Possible Dust Aerosols Semi-Direct Effect on Cloud Water Path over East Asia." Geophysical Research Letters 33 (19):L19802. https://doi.org/10.1029/2006GL026561.**

We agree with the reviewer that the manuscript and the discussion would improve by including the recommended references. References are added in the manuscript.

**3. Line 10: "airborne mineral dust is considered a significant atmospheric aerosol contributor", should be corrected into "...considered as a..."**

Corrected.

**4. Page 2, Line 29-32: "although passive satellite sensors provide information on the column properties of aerosols with adequate spatial and temporal resolution, they are bound to certain limitations, the major limitation being the lack of information on the three-dimensional distribution**

**(vertical profile) of aerosols in the atmosphere, an important information for the assessment of the aerosols radiative forcing on climate as well as their contribution as IN and CCN (IPCC 2013)."** This sentence is too long to understand it means, please rewrite it.

The long sentence was re-written, in order to be easier to understand: "Although passive satellite sensors provide information on the column properties of aerosols with adequate spatial and temporal resolution, they are bound to certain limitations. The major limitation is the lack of aerosol information on the three-dimensional distribution (vertical profile), which consist of an important parameter for the assessment of the aerosols radiative forcing on climate as well as their contribution as IN and CCN (IPCC 2013).".

**5. Page 4, line 10: I think you should delete this words: "in order to discriminate the detected atmospheric features types into subtypes", because we have known the goal of the classification algorithm before this sentence.**

The authors are of the opinion that this section consists a methodology bridge to the pure-dust product, between the CALIPSO algorithm and aerosol subtype classification and the decoupling of the pure-dust component from the classified as dust and polluted dust aerosol layers by CALIPSO . Towards this goal and since the methodology section largely is based on the classification algorithm, the authors considered that these lines should not be deleted, but under consideration of the recommendation of the reviewer they are modified to:
"The Level-2 (L2) product consists the high-level quality products. More specifically, CALIPSO L2 algorithm classifies the detected layers into characteristic classes (Vaughan et al., 2009), namely into clear air, cloud, aerosol, stratospheric, surface, subsurface, totally attenuated or invalid feature types. The classification algorithm (Omar et al., 2009) utilizes the depolarization ratio and the magnitude of the attenuated backscatter signal, the height of the aerosol layers and the characteristics of the Earth's surface along the CALIPSO footprint (desert, ocean, snow/ice) in order to discriminate the detected atmospheric features types into subtypes".

**6. Page 4, line 14: the "2ox5o grid resolution" should be corrected into "2°x5°".**

Corrected.

**7. Page 4, line 17: "1ox1o" need to be corrected. The whole paper should be checked again.**

Corrected. The whole paper was checked again.

**8. Page 6, line 18: since you have said that the daytime minimum and nighttime minimum, what does the "minimum detectable AOD of 0.005" mean?**

According to the reviewer's recommendation the text is corrected to: "Regarding the uncertainties of the products, CALIOP L2 V3 is characterized by daytime minimum detectable backscatter of $0.0017\pm0.0003$ km$^{-1}$sr$^{-1}$, nighttime minimum detectable backscatter of $0.0008\pm0.0001$ km$^{-1}$sr$^{-1}$ and AOD of 0.005 (based on the minimum CALIOP 532 nm channel detection sensitivity, Winker et al., 2009)".

**9. Page 4, line 24: what does the "SAMUM" mean? Please write the full name.**

The text is modified according to the reviewer's recommendation: "During the SAharan Mineral dUst experiMent (SAMUM) 1 and 2 campaigns Saharan dust ...".

**10. Page 4, line 51: in this paragraph you introduce the methods of distinguish pure dust and non-dust. However, I still don't know the differences of the CALIPSO product of dust and polluted dust with the pure dust and non-dust. Since we can directly derive the dust extinction coefficient and profiles from the product, why don't you use it? And what about merits of the method to select the pure dust? What's the differences of the pure dust and dust products directly from CALIPSO L2?**

The CALIPSO V3 aerosol classification algorithm classifies the detected aerosol features as marine, dust, clean continental, polluted continental, polluted dust and smoke (Omar et al., 2009). Typical dust particle depolarization ratio values measured with lidars in field campaigns around the globe show values between 0.27 and 0.35 at 532 nm. Furthermore, the measurements show little variation independently of the source region, (e.g., Ansmann et al., 2011; Sakai et al., 2000; Liu et al., 2008b; Freudenthaler et al., 2009; Groß et al., 2011; Burton et al., 2013; Groß et al., 2013; Groß et al., 2015; Illingworth et al., 2015). Based on the dust depolarization ratio, a methodology has been established to discriminate the pure-dust component from mixtures of dust and non-dust aerosol layers (Tesche at al. 2009). In this methodology both the CALIPSO dust and polluted dust aerosol types are treated as mixtures of dust aerosols and non-dust aerosols. The methodology is applied and the final CALIPSO pure-dust product (the pure-dust component of the dust/polluted dust layers of CALIPSO) (Amiridis et al., 2013) are available to perform CALIPSO climatological studies (Marinou et al., 2017) and to develop interesting dust-related products (LIVAS-Amiridis et al., 2015).

**11. Page 5, line 30: please check this sentence of "The seasonal zonal distribution of the climatological and conditional dust extinction coefficient (Mm$^{-1}$)". If it's right to explain it.**

The reviewer is right, this was an editing error by the authors. The author's intension was to implement typographical symbol in order to introduce a list of CALIPSO products that would be used in the study and accordingly extensively discussed. Omitting the typographical symbol resulted in much confusion and we apologize for this mistake. The symbols have been restored, the list is clarified along with the sentence.

**12. Page 5, line30: I want to know whether the climatological dust extinction coefficient means the aerosol extinction without dust extinction coefficient since you write this sentence "This is accomplished by setting the dust extinction coefficient value of 0 km-1, for observations with non-dust aerosols". And the conditional dust product only has the dust extinction coefficient.**

The climatological extinction coefficient is computed by setting the extinction coefficient value of the non-dust aerosols to 0 km$^{-1}$, when averaging the profiles over a grid. The authors agree with the reviewer that this part of the manuscript was not clear, therefore it is re-written as follows:
"The climatological dust product is a measure of the average dust load over a geographical domain and is computed acknowledging only the contribution of the dust component in the atmosphere. Technically, this is accomplished by setting the extinction coefficient value of the non-dust aerosols to 0 km$^{-1}$, when averaging the profiles over a grid. The dust climatological product can be used for studies related to the contribution of dust to the total aerosol load over a period of time. In addition, the climatological dust product can be used in the evaluation of models related to dust transport and to radiative transfer models, in studies of dust-related physical processes (dust transport dynamics, CCN, IN), to investigate the effect of dust aerosols on ecosystems (dust deposition into the oceans) and to determine the dust aerosol load over highly industrialized and densely populated regions.
The conditional dust product is a measure of the average intensity of dust load over a geographical domain and is based explicitly on the dust profiles, hence ignoring completely non-dust aerosols. Technically, this is accomplished by setting the extinction coefficient value of the non-dust aerosols to

not-a-number (NaN), when averaging the profiles over a grid. The conditional dust product is related to the intensity of the dust events."

**13. Page 7, line 33: what does the "N. China" mean?**

Corrected to: "Over N. China, for latitudes northern than 35° N, a similar pattern with respect to the features of dust contribution to the total aerosol load due to the dust aerosol emitted from the Taklimakan and Gobi deserts are observed".

**14. Page 8, line 11: from the figure 3, the differences of dust frequency in the four seasons are not clear, and the minimum in Fig. 3a is not obvious.**

Both the scale and the colormap of the dust frequency, CoM and TH are modified, according to the suggestion by the reviewer. Please see the figures below, before and after the adaptation of the figures.

**Before**

[Figure]

Fig. 3: Spatial distribution of dust occurrence [%], climatological pure-dust CoM (Center of Mass) and dust TH (Top Height) in km a.g.l., for each season over the domain between 65°-155° E and 5°-55° N and for the period 01/2007-12/2015.

**After**

Dust Cases / Overpasses          Dust Center of Mass          Dust Top Height

[Figure]

Fig. 3: Spatial distribution of dust occurrence [%], climatological pure-dust CoM (Center of Mass) and dust TH (Top Height) in km a.g.l., for each season over the domain between 65°-155° E and 5°-55° N and for the period 01/2007-12/2015.

**15. Page 8, line 39: please explain the pattern of the dust transport since you said "however, the pattern reverses (Fig. 3i)"**

Corrected to: "During MAM, dust particles emitted from the Taklimakan and Gobi deserts are transported over C. China and the Pacific Ocean, while at the same time significant long-range transport of dust aerosols emitted from Thar Desert is not-observed (Fig. 3f). During JJA, however, the pattern reverses, with longer range of dust particles transported from Thar Desert over the Indian Peninsula, the Arabian Sea and the Bay of Bengal, while no significant dust transport of dust aerosol emitted from Taklimakan Desert is observed (Fig. 3i)."

**Response to Anonymous Referee #2**

**The paper by Proestakis et al. presents a 9-year climatology derived with mainly CALIOP measurements of the aerosol (dust) conditions over East and South Asia. The main focus is set on dust distribution but also non-dust aerosol is discussed. These novel results are thus of interest for atmospheric research and give a good overview of the dust distribution in this part of the world. Therefore in principle the paper is of interest for publication in ACP, however I recommend major revisions before it can be published. This is further explained below.**

The authors would like to thank the referee for the interesting and at the same time substantial constructive comments and suggestions. We tried, and did our best, to incorporate the proposed changes and corrections in the revised manuscript, aiming to the improvement of the presented paper. Following, you will find our responses that are addressed to the Editorial board and the reviewers as well.

**Major comments**

**In my opinion the naming of the study area is misleading. Even though South-East Asia may not be a protected phrase, many people have a different understanding concerning the region called this way. See for example:**
**https://en.wikipedia.org/wiki/Southeast_Asia**
**Therefore, I recommend to find a better name for the study area (e.g. South and East Asia or whatever) to avoid confusion and change the title and text accordingly.**

According to the reviewer's recommendation the name of the study area is changed from "Southeast Asia" to "South and East Asia", both in the title and the text.

**The difference between the climatological and conditional dust product needs be more discussed in both, the methodology section (I just understood the difference when reading Marinou 2017 but as this is essential it should be explained more explicitly here), but also in the result section. The reader is left alone with contradictory statements, like for example the dust top height which seems to be completely different between the two products. Therefore it should be clearly discussed:**
**(1) Which product can be used for which purpose?**
**(2) Does it make sense to use this two different products, if yes why and for what?**
**(3) What we can learn from the two products presented here with respect to South and East Asia.**

Both the "Data and Methodology" and the "Results" Sections have been revised and re-written according to the recommendation of the reviewer. To be more specific, (1) and (2) have been re-written and extended as follows:
The seasonal zonal distribution of the climatological and conditional dust extinction coefficient ($Mm^{-1}$). The climatological dust product is a measure of the average dust load over a geographical domain and is computed acknowledging only the contribution of the dust component in the atmosphere. Technically, this is accomplished by setting the extinction coefficient value of the non-dust aerosols to 0 km-1, when averaging the profiles over a grid. The dust climatological product can be used for studies related to the contribution of dust to the total aerosol load over a period of time. In addition, the climatological dust product can be used in the evaluation of models related to dust transport and to radiative transfer models, in studies of dust-related physical processes (dust transport dynamics, CCN,

IN), to investigate the effect of dust aerosols on ecosystems (dust deposition into the oceans) and to determine the dust aerosol load over highly industrialized and densely populated regions.

The conditional dust product is a measure of the average intensity of dust load over a geographical domain and is based explicitly on the dust profiles, hence ignoring completely non-dust aerosols. Technically, this is accomplished by setting the extinction coefficient value of the non-dust aerosols to not-a-number (NaN), when averaging the profiles over a grid. The conditional dust product is related to the intensity of the dust events.

In addition to the above a Flowchart is provided according to the reviewer's suggestion in the end of the ""Data and Methodology" Section (comment 4.3).

(3) is additional included. In general the findings are summarized in "Summary and conclusions" Section, which is re-written.

**Conclusion: The current conclusion is not very informative. Thus, it should be really overworked to highlight new things and discuss what lessons have been learned, i.e. what are new results or newly gathered knowledge or does your study just confirm former studies etc...**

The authors agree with the reviewer, the conclusion section was re-written in order to provide more information and highlights of the study. Below we provide a part of the conclusion which was vastly rephrased and extended: "In this work, CALIPSO is used to provide a multiyear 4-D climatology of desert dust aerosols over South and East Asia at a spatial resolution of 1ox1o deg grids. An optimized dust aerosol product, developed using CALIOP backscatter and particle depolarization ratio, along with a regional correction on dust lidar ratio suitable for Asian dust is used. The optimized product is utilized to provide the horizontal and vertical distribution along with the temporal evolution of dust aerosols over a 9-year period (01/2007-12/2015). Regarding the horizontal distribution of [1]erosol Optical Depth (AOD), Dust Aerosol Optical Depth (D_AOD) and Non-Dust Aerosol Optical Depth (Non-Dust AOD), our analysis shows similar patterns between all four seasons, although the magnitude of the observed features varies with season. High values of Non-Dust AOD are consistently observed over the heavily industrialized and densely populated regions of China and India (Non-Dust AOD > 0.5). In addition to the anthropogenic densely populated areas of South and East Asia, the major sources of dust aerosols, namely the Taklimakan, Gobi and Thar Deserts are clearly mapped through the systematic high D_AOD values throughout the year. The magnitude though of the D_AOD observed features is subject to high seasonality, ranging between D_AOD 0.2 during winter and higher than 0.6 during spring and summer seasons. Maximum activity of Gobi and Taklimakan deserts is observed during spring, while the highest activity of Thar Desert is during summer. The seasonality of the dust transport pathways is additionally well-captured. Dust transport over the Indian Peninsula is more pronounced during spring and summer, while over China similar patterns of a persistent dust aerosol background is evident throughout the year, with a peak during spring when the dust transport across the Pacific Ocean is at its maximum.

Regarding the vertical distribution of dust aerosols, the Center of Mass (CoM), Top Height (TH) and the mean Dust Extinction Coefficient profiles (Climatological and Conditional) are [2]nplemented to provide, together with the horizontal distribution, the full three-dimensional structure of dust aerosols and the atmospheric dust transport pathways over the entire South and East Asia. Based on the synergy of CoM, TH and the CALIPSO dust extinction profiles two distinct dust transport pathways over South and East Asia are observed: a [3]he Trans-Pacific belt between 25o and 45o N and a second one, extending from Thar Desert towards the Bay of Bengal and the Arabian Sea. Both zones of dust transport are subject to high seasonality. Highest dust aerosol transport from the Taklimakan Desert towards the Pacific Ocean is observed during spring, while dust aerosol transport from the desert of Thar and across the Indian Subcontinent is more pronounced during summer.

Regarding the temporal evolution of AOD and D_AOD between 01/2007 and 12/2015, the analysis showed statistically significant positive short-term AOD trends over the Indian Peninsula (0.01 yr-1), NW China (0.007 yr-1) and E China (0.01 yr-1), whereas our study shows negative short-term AOD trends over SE China (-0.007 yr-1). [4]ALIPSO positive AOD trends are found over the broader central and eastern Indian Peninsula (0.01 yr-1). The CALIOP observed trends between 01/2007 and 12/2015 are generally in qualitative agreement with the derived MODIS AOD trends over large domains of South and East Asia, although the short-term trends disagree over specific regions. The CALIOP and MODIS trends though are interpreted and compared with caution, since the samples of the datasets are non-uniform. "

**Number: 1     Author: Referee     Subject: Comment on Text     Date: 20.12.2017 11:26:40**
As AOD can be also measured from other satellites, I wonder if you should highlight that such precise dust and non-dust distinction is only possible from lidar with polarization capabilities, i.e. only with Calipso. Or i am wring and one could do the same with e.g. MODIS?

**Number: 2     Author: Referee     Subject: Comment on Text     Date: 20.12.2017 11:24:21**
evaluated?

**Number: 3     Author: Referee     Subject: Cross-Out  Date: 20.12.2017 11:26:50**

**Number: 4     Author: Referee     Subject: Cross-Out  Date: 20.12.2017 11:28:09**

**Please check spelling and grammar intensively again. There are many sentences which are fractals, i.e. words are missing. Furthermore, many commas are missing, considering the bunch of co-workers this should be no problem.**

5     The authors have gone through the entire manuscript again to check for spelling and grammar again. At this point the authors would like to thank the reviewer once more for the substantial contribution towards the direction of improving the overall manuscript.

**Minor comments**

**Page 1.1:** According to the reviewer's recommendation the name of the study area is changed from
15     "Southeast Asia" to "South and East Asia", both in the title and the text.
**Page 1.2:** According to the reviewer's suggestion, two sentences are used to simplify the initial sentence: "To distinguish desert dust from total aerosol load we apply a methodology developed in the framework of EARLINET (European Aerosol Research Lidar Network). The methods involves the use of particle linear depolarization ratio and updated lidar ratio values suitable for Asian dust, from multiyear CALIPSO
20     observations (01/2007-12/2015)."
**Page 1.3:** Wavelength is included: "532 nm".
**Page 1.4:** Suggestion is included.

**Page 2.1:** Suggestion is included, the text is modified accordingly.
25     **Page 2.2:** The text is corrected: from "Major dust Asian" to "Major Asian dust".
**Page 2.3:** According to the suggestion the text is modified: from "CALIOP measures total attenuated backscatter signals at …" to "CALIOP measures total attenuated backscatter at …"

**Page 3.1:** According to the suggestion the text is rephrased: "Using this classification they either did not
30     take into consideration the dust component 1 f the classified as 2 polluted dust aerosol subtype, or they defined as "dust" both the dust and polluted dust aerosol subtypes (hence including the non-dust component of polluted dust)".
**Page 3.2:** Suggestion is included.
**Page 3.3:** Suggestion is included: "… this new pure dust …".
35     **Page 3.4:** Suggestion is included: "… pure dust product …".
**Page 3.5:** The text is corrected: from "laser" to "lidar".
**Page 3.6:** Since according to the reviewer the sentence was not clear the text is rephrased to: "CALIOP transmits linear polarized light, while a telescope of 1 m diameter collects the backscatter component backscattered by the atmosphere.".

**Page 4.1:** The text is modified according to the reviewer's recommendation and the different feature type classes are included: "The Level-2 (L2) product 5 onsists the high-level quality products. More specifically, CALIPSO L2 algorithm classifies the detected layers into characteristic classes (Vaughan et al., 2009), namely into clear air, cloud, aerosol, stratospheric, surface, subsurface, totally attenuated or
45     invalid feature types."
**Page 4.2:** The manuscript is cross-checked and the aerosol subtype classification scheme is not explained before this part in the data and methodology section. Part of the paragraph though is rephrased: "The classification algorithm (Omar et al., 2009) utilizes the depolarization ratio and the magnitude of the attenuated backscatter signal, the height of the aerosol layers and the characteristics of the Earth's
50     surface along the CALIPSO footprint (desert, ocean, snow/ice) in order to discriminate the detected

Number: 1        Author: Referee        Subject: Cross-Out  Date: 20.12.2017 11:31:22

Number: 2        Author: Referee        Subject: Cross-Out  Date: 20.12.2017 11:31:26

Number: 3        Author: Referee        Subject: Cross-Out  Date: 20.12.2017 11:34:13

Number: 4        Author: Referee        Subject: Inserted Text        Date: 20.12.2017 11:34:11
s are

atmospheric features types into subtypes. The atmospheric features types classified as aerosols are further distinguished into specific aerosol subtypes (Clean Marine, Dust, Clean Continental, Polluted continental, Polluted Dust and Smoke)."

Page 4.3: According to the suggestion the following Flowchart, diagrammatic representation from the CALIPSO data to the Pure-Dust product and the Climatological/Conditional Dust products used in the study is added to the end of the "Data and Methodology" Section.

[Figure]

Fig. 2: Flowchart of the CALIPSO Pure-Dust, Conditional Dust Extinction Coefficient and Climatological Dust Extinction Coefficient products.

Page 4.4: The text is corrected: from "… with cloud observations are filtered from …" to "… with cloud observations are filtered out from …".

Page 4.5: According to the suggestion the text is modified: from "… value of the pure dust component, $\beta_\perp$ …" to "… value of the pure dust component in the aerosol mixture, $\beta_\perp$ …"

Page 4.6: According to the recommendation by the reviewer the following two references are added: "Omar et al., 2009".

Page 5.1: The authors agree with the reviewer that the paragraph in the beginning of the "Data and Methodology" section is confusing, therefore the entire paragraph was moved to the end of the "Introduction Section" at the part of the description of the study domain.

Page 5.2: According to the reviewer's recommendation the name of the study area is changed from "Southeast Asia" to "South and East Asia".

Page 5.3: The backscatter coefficient.

Page 5.4: The text is corrected: from "extinction coefficient" to "backscatter coefficient".

Number: 1    Author: Referee    Subject: Comment on Text    Date: 20.12.2017 11:39:58

That's not true. delta_p is the particle depolarization ratio containing all particles in the mixture. The depolarization value of the pure dust component you assume with delta_1 in formula (2) and use this for separation.

**Page 5.5:** The reviewer is right, this was an editing error. The author's intension was to implement typographical symbol in order to introduce a list of CALIPSO products that would be used in the study and accordingly discussed. Omitting the typographical symbol resulted in much confusion to both the reviewers and we apologize for this mistake. The symbols have been restored and the list is clarified along with the sentence.

**Page 5.6:** Done (major comment and comment 4.3).

**Page 5.7:** Done (major comment and comment 4.3).

**Page 5.8:** According to the reviewer's recommendation the sentence is modified to include which product: "Validation of the pure dust aerosol product against …".

**Page 5.9:** According to the reviewer's recommendation the sentence is modified to include the domain: "… observations over northern Africa and Europe show …".

**Page 5.10:** The sentence did not make sense since it is part of a list which was omitted. Corrected through the introduction of the list and the typographical symbols (comment 5.5).

**Page 6.1:** Full stop added.

**Page 6.2:** Wavelength is included: "532 nm".

**Page 6.3:** Discuss AOD uncertainty is approximately 100% close to the surface (Marinou et al., 2017).

**Page 6.4:** According to the suggestion from the reviewer the sentence "Additional, uncertainty which propagates into the D_AOD product is introduced due to the depolarization ratio of the non-dust aerosols, coupled into the polluted-dust aerosol subtype." is rephrased to: "In addition, as it is already mentioned, both aerosol types classifies by CALIPSO as dust or polluted dust are a mixture of a dust component and a non-dust component. Thus another source of uncertainty in the decoupling of the dust component from the total aerosol load is the lack of information regarding the non-dust component in the aerosol mixture, due to the low depolarization ratio values of the non-dust aerosol subtypes (Omar et al., 2009)."

**Page 6.5:** According to the suggestion from the reviewer the sentence "… the seasonal approach is selected …" is rephrased to: "… in this section we present and discuss the horizontal distribution of aerosols and dust over South and East Asia per season."

**Page 6.6:** According to the suggestion from the reviewer the entire paper is homogenised. "ΔOD" is replaced to "Non-Dust AOD".

**Page 7.1:** Satellite-based remote sensing, both passive and active, is highly sensitive to the presence of clouds. Regarding CALIOP, the nadir-viewing lidar measurements and orbital characteristics of CALIPSO result in a low frequency of overpasses over each region and consequently the significant fewer observations with respect to passive sensors. Therefore, in order to provide meaningful climatologies on a regional scale, long-term and multiyear CALIOP observations are required. Even though over specific regions which are characterized of extensive cloud coverage of dense cloud (Bay of Bengal, Indonesia, N. Pacific Ocean), the number of observations in the sample is even sparser, due to attenuation in dense clouds. Therefore rare regional events are not well captured over regions of extensive cloud coverage, resulting in weighted values toward regions of less extensive cloud coverage (Winker et al., 2013).

**Page 7.2:** Indochina is a political and geographical correct term. Indochina, originally Indo-China, is a geographical term originating in the early nineteenth century and referring to the continental portion of the region now known as Southeast Asia. The name refers to the lands historically within the cultural influence of India and China, and physically bound by the Indian Subcontinent in the west and China in the north. It corresponds to the present-day areas of Myanmar, Thailand, Laos, Cambodia, Vietnam, and (variably) peninsular Malaysia and Singapore. See for example: https://en.wikipedia.org/wiki/Indochina

**Page 7.3:** According to the suggestion from the reviewer the sentence is modified to: "Over China similar geographical patterns in the horizontal distribution of aerosols are evident between all four seasons, with larger Non-Dust AOD values over the major sources of anthropogenic aerosols [1]Beijing, Shanghai,

Number: 1        Author: Referee        Subject: Cross-Out  Date: 20.12.2017 11:54:28

⊡uangzhou, Chongqing, Wuhan) such as urban clusters (Beijing, Shanghai, Guangzhou, Chongqing, Wuhan) (Kourtidis et al., 2015) and high D_AOD values over the deserts of Taklimakan and Gobi (Che et al. 2014, 2015)".

**Page 7.4:** Done (comment 7.3).

**Page 7.5:** Done (comment 7.3).

**Page 7.6:** Corrected: "… high D_AOD values …".

**Page 7.7:** For example to the north of the plateau of Tibet, during the period between March and May the strong surface winds which develop over the Mongolian Plateau create favourable mechanisms of extreme dust events (Bory et al., 2003; Yu et al., 2008). This feature is evident throughout the year, although more pronounced during spring. By activation of the deserts the authors mean the creating of favourable conditions for dust generation and injection in higher altitude in the atmosphere.

**Page 7.8:** The sentence is rephrased to: "Over China, for latitudes northern than 35o N, a similar pattern with respect to the features of dust contribution to the total aerosol load due to the dust aerosol emitted from the Taklimakan and Gobi deserts are observed. More specific, a persistent dust aerosol background is evident during all seasons, with a peak during MAM (Fig. 3f)."

**Page 7.9:** Rephrased (comment 7.8).

**Page 7.10:** This is in line with previous studies, reporting rare events of dust transport over Himalayas (Huang et al., 2007; Liu et al., 2008b; Yumimoto et al., 2009). The region to the North of Taklimakan, Gobi and Mongolia is also characterized by low values of AOD and D_AOD, except during MAM (Fig. 3f, h). The high dust aerosol load observed to the east of the major dust aerosol source of Taklimakan (D_AOD values greater than 0.3) and the high percentage of D_AOD with respect to the total AOD indicate a strong eastward transport of both dust (Fig. 3f) and anthropogenic aerosols (Fig. 3g). The paragraph is modified accordingly.

**Page 8.1:** Both suggestions are implements, the a.g.l. is added to the figure caption and the common nomenclature are used.

**Page 8.2:** According to the reviewer's recommendation the name of the study area is changed from "Southeast Asia" to "South and East Asia".

**Page 8.3:** According to the recommendation by the reviewer we have included some numbers. More precisely the following part is added: "Lower frequencies of dust occurrence, which still exceed 70%, are also evident over east China and south-eastern India. Conversely over Indochina and Indonesia the occurrence of dust is particularly low, especially during summer (Fig. 4g) and autumn (Fig. 4j). To be more specific, values of dust occurrence percentage between 50% and 60% over Thailand and Cambodia, 40% to 60% over Laos and Vietnam, ~60% over SE China, and lower than 40% over Malaysia and Philippines are observed, during JJA and SON.".

**Page 8.4:** Comma added.

**Page 8.5:** "s" deleted.

**Page 8.6:** The authors are not sure in which paper the reviewer is referring to, though we suspect that the differences most probably are related to the definition of the TH and CoM, probably to differences in the above ground level or above sea level reported profiles, maybe to different seasonality, sensor detection limits, differences in the techniques applied, to different samples. There are just many factors that may result in the different observations.

**Page 8.7:** The sentence is rephrased according to the suggestion of the reviewer, since the meaning was not clear. The sentence is rephrased from: "In addition, the observed gradient in the horizontal distribution of D_AOD values between the sources and the Pacific Ocean (Fig. 3f), parallel to the ubiquitous dust layer and the high dust TH (Fig. 4f), are an indicator of the longer range of transport of lower concentration of dust particles." to "In addition, a decreasing west-to-east D_AOD gradient is observed over N. China, between the dust sources over Taklimakan and Gobi and the Pacific Ocean (Fig.

Number: 1   Author: Referee   Subject: Cross-Out Date: 20.12.2017 11:54:28

3f). The decreasing gradient of TH is less pronounced during MAM, when dust aerosol are injected as high as 10 km height (a.s.l.) and transported longer distances over the Pacific Ocean (Fig. 4f).".

**Page 8.8:** Comma added.

**Page 8.9:** Since according to the reviewer the meaning was not clear, the paragraph was rephrased accordingly: "during MAM, dust particles emitted from the Taklimakan and Gobi deserts are transported over C. China and the Pacific Ocean, while at the same time significant long-range transport of dust aerosols emitted from Thar Desert is not-observed (Fig. 4f). During JJA, however, the pattern reverses, with longer range transport of dust particles from Thar Desert over the Indian Peninsula, the Arabian Sea and the Bay of Bengal, while no significant dust transport of dust aerosol emitted from Taklimakan Desert is observed (Fig. 4i).".

**Page 8.10:** Sentence corrected (comment 8.9).

**Page 8.11:** "larger" is replaced by "higher" according to the recommendation by the reviewer.

**Page 8.12:** Adapted (comment 27.1).

**Page 8.13:** Comma deleted.

**Page 8.14:** "into" is replaced by "over" according to the recommendation by the reviewer.

**Page 9.1:** "the" deleted.

**Page 9.2:** Sentence is modified accordingly: "… provided in Table 1".

**Page 9.3:** The pure dust component is of interest to regions where dust aerosols are present. In the domain of South and East Asia, such domains are East China and India. East China is affected from dust aerosol transport from the Taklimakan and Gobi, while India is affected from dust transport from Thar Desert and the Arabian Peninsula. The significance of dust load is even larger due to the anthropogenic emissions of the densely populated provinces of China and the regions of India.

**Page 9.4:** Initially we had a column of the study domain, encompassing the different domains. In general, after initial submission, we have tried to move the individual figures as close to each other as possible, otherwise the individual plots won't exceed stamp size in the final paper and the reader won't be able to recognise a thing. Towards this need, we had to omit as much redundant information from the initial submission as possible, including the map plots which indicated the bands of interest, since the reader can take this information from Figure 1.

**Page 9.5:** Corrected: "The continuous and dashed lines correspond to the average elevation of the surface level and to the average maximum elevation respectively.".

**Page 9.6:** This threshold is arbitrarily selected to filter out low dust aerosol extreme cases scenarios in order to limit the influence of rare events on climatology. To be clearer, the following was added: of the pure dust climatological extinction coefficient, arbitrarily selected, in order to avoid presenting extreme rare events in high altitudes at the same time with climatological values close to the surface level

**Page 9.7:** Indeed the sentence "Regarding the Taklimakan Desert, this region is a very prolific arid area encompassed by Tarim Basin" is clarified to "Taklimakan Desert, consists a very arid area encompassed by Tarim Basin".

**Page 10.1:** Indeed, the reviewer is right, the sentence is modified accordingly: from "during MAM lofted of dust extinction coefficient ..." to "… during MAM dust aerosol layers are detected …".

**Page 10.2:** According to the opinion of the authors, the CoM should not be plotted together with the climatological dust extinction coefficient, and the reason is threefold. First of all, the purpose of the climatological dust product is to answer the question, which is the aerosol load over a specific region on a climatological basis, while the purpose of the CoM is to provide the climatological center of mass of the average dust profile. A rare example is related to a region where dust is never present. Over such a region the climatological dust profile is a well-defined profile of zeros. On the contrary, by mathematical definition, the CoM over such profile cannot be computed (NaN). The second reason is related to the representativeness. An example over the study domain of South and East Asia is the Himalayas orographic barrier. Extreme and very rare events of dust transport over Himalayas have been reported.

Number: 1    Author: Referee    Subject: Inserted Text    Date: 20.12.2017 12:01:48
over

The CoM plotted together with the climatological dust extinction coefficient profiles would create to the reader a misleading view of frequent dust transport over this region, due to the lack of the information of how many events are used to compute the profile and the CoM. The third reason is related to the fact that the CoM is always defined above ground level. Due to the complex surface orography of South and East Asia, the mean surface elevation of the used regions sometimes may be above the CoM. For example Tarim Basin, the basin that encompasses the Taklimakan Desert has an elevation approximately 2 km higher than the Taklimakan desert elevation. Thus during the non-active seasons the dust CoM would be below or very close to the mean Surface elevation of the region.

**Page 10.3:** Indeed, the sentence is modified accordingly from "... attributed the gravitational settling of aerosols and to dry and wet deposition (Colarco et al., 2003)." to "... attributed to both dry and wet deposition processes that remove dust aerosol from the atmosphere (Colarco et al., 2003).".

**Page 10.4:** Indeed, the reviewer is right, the sentence is modified accordingly: from "MAM a lofted layer of dust climatological extinction coefficient up to 25 Mm$^{-1}$ is observed ..." to "…MAM a lofted layer of dust aerosols that yields climatological extinction coefficient up to 25 Mm$^{-1}$ is observed …".

**Page 10.5:** The Climatological Total Extinction Coefficient close to the surface over the densely populated regions of East China, especially during MAM when a large component of the AOD is related to dust aerosols transported from Taklimakan and Gobi deserts is as high as 200 Mm$^{-1}$. For more information: de Leeuw, G., Sogacheva, L., Rodriguez, E., Kourtidis, K., Georgoulias, A. K., Alexandri, G., Amiridis, V., Proestakis, E., Marinou, E., Xue, Y., and van der A, R.: Two decades of satellite observations of AOD over mainland China, Atmos. Chem. Phys. Discuss., https://doi.org/10.5194/acp-2017-838, in review, 2017.

**Page 11.1:** The difference between climatological and conditional is improved and adapted and already discussed.

**Page 11.2:** "coefficient" deleted.

**Page 12.1:** Since according to the reviewer it was not clear through the sentence where the features of dust transport pathways are observed in Figure 6, the sentence is modified accordingly: "To the north and east of the Tibetan Plateau two distinct eastward pathways of dust transport are evident: (1) a northern flow that propagates towards the Yellow Sea and the Pacific Ocean (Uno et al., 2009) (Fig. 6f, g) and (2) a southern flow that occurs over central China (Kuhlmann and Quaas, 2010) (Fig 6k, o)."

**Page 12.2:** Corrected. "Both transport pathways are observed at the middle and upper troposphere …".

**Page 12.3:** Indeed the authors agree with the reviewer that the paragraph was not written in a clear way to help the reader to navigate along the figures and easily absorb the information provided. Thus the entire paragraph was rewritten: "To the north and east of the Tibetan Plateau two distinct eastward pathways of dust transport are observed: (1) a northern flow that propagates towards the Yellow Sea and the Pacific Ocean (Uno et al., 2009) and (2) a southern flow that occurs over central China (Kuhlmann and Quaas, 2010). The northern flow is mostly evident during winter (Fig. 3d), while the southern transport pathway over C. China is more prominent during spring (Fig. 3h). Figure 6 provides information on the vertical distribution and depth of the two dust transport pathways. Both transport pathways are observed at the middle and upper troposphere, indicated by dust conditional coefficient values as high as 20 Mm$^{-1}$, observed at altitude up 10 km a.s.l. Another noticeable feature is that the vertical intensity of the transported dust aerosol plumes is subject to high spatial and seasonal variability. Decreasing values of both dust aerosol climatological and conditional values are observed with increasing distance from the dust sources of Taklimakan and Gobi deserts towards and over the Pacific Ocean.".

**Page 12.4:** Comment 12.3

**Page 12.5:** Comma deleted.

**Page 12.6:** Comment 12.3

**Page 12.7:** The reviewer is right. "(as discussed in Section Data and Methodology)" is added.

**Page 12.8:** According to the different particle depolarization values, a mean value of 0.03 value is used for the non-spherical aerosol types. This introduces uncertainties since the 0.03 is a mean value, ±0.01.
**Page 12.9:** "particle" added.
**Page 12.10:** Indeed the sentence is rephrased: "Figure 7 shows the vertical, horizontal and seasonal variability of the average particle depolarization ratio of the cases classified by CALIOP as dust or polluted dust aerosol subtypes based on nine years of CALIPSO observations (01/2007-12/2015) and for five zones of 10° latitudinal interval, between 5° and 55° N.".
**Page 12.11:** The misleading sentence is rephrased according to the reviewer's recommendation from "In general, to the north of Himalayas (Fig. 7e-h), a structure of three different height ranges is evident." to: "In general, to the north of Himalayas, low values of particle depolarization ratio are observed close to the surface, while particle depolarization ratio increases with increasing height (Fig. 7e-h).".

**Page 13.1:** Rephrased to: "Over Thar Desert the average particle depolarization ratio of cases classified as dust or polluted dust by the CALIPSO classification algorithm yield average depolarization values greater than 25% throughout the year".
**Page 13.2:** Height in Fig. 4 are defined above ground level while at Fig. 5, 6 and 7 characteristic are above sea level.
**Page 13.3:** Comma added.
**Page 13.4:** The sentence is deleted after cross-checking.
**Page 13.5:** The entire manuscript is homogenised accordingly.

**Page 14.1:** Over the manuscript frequently the authors have discussed the horizontal and vertical distribution of AOD and dust aerosol respectively. Both MODIS Aqua and CALIPSO CALIOP are utilized. Both sensors utilize different algorithms for cloud screening for cloud fraction and extensive cloudiness prevent retrievals over specific domains (Bay of Bengal and monsoon period). These figures, although not extensively discusses, are in the manuscript for completeness reason, for a reviewer to have a full overview of the datasets utilized.
**Page 14.2:** According to the reviewer's recommendation the sentence is rephrased from "Strong statistical increase is observed by MODIS over the Arabian Sea (0.01 $yr^{-1}$), while not statistical significant positive AOD550nm trends are present over the Bay of Bengal (0.002 $yr^{-1}$)." to "MODIS shows not statistical significant AOD increasing trends of the order of (0.002 $yr^{-1}$) over the Bay of Bengal and strong positive statistical significant trends over the Bay of Arabian Sea (0.01 $yr^{-1}$). The strongly increasing AOD trend over the Arabian Sea though is not corroborated by CALIOP observations.".
**Page 14.3:** The abbreviation is removed according to the reviewer's suggestion.

**Page 15.1:** Indeed, the reviewer is right. Based on CALIOP both the horizontal and vertical distribution of aerosol can be studied.
**Page 15.2:** According to the reviewer's recommendation the abbreviations are explained one last time in the beginning of the "Summary and Conclusions" section. To be more specific the following phrase is modified: "Our analysis shows similar patterns in the horizontal distribution of Aerosol Optical Depth (AOD), Dust Aerosol Optical Depth (D_AOD) and Non-Dust Aerosol Optical Depth (Non-Dust AOD) between all four seasons ...".
**Page 15.3:** The authors mean, the range of the observations, how high/low the observed values are.
**Page 15.4:** Corrected.
**Page 15.5:** According to the reviewer's recommendation the sentence is rephrased: "The CoM, TH and the mean extinction coefficient profiles are implemented in order to provide the vertical dust aerosol distribution, and together with the horizontal distribution of AOD and D_AOD to provide in the end the three-dimensional structure of dust aerosol over South and East Asia and the atmospheric dust aerosol transport pathways.".

**Page 15.6:** "The Summary and Conclusions" Section is adapted according to the reviewer's suggestion (major comment).

**Page 15.7:** The sentence is rephrased. Actually the entire "Summary and Conclusions" Section is modified. The Angstrom Exponent and the spectral dependence of AOD is crucial since it can be used as a fingertip for identifying coarse aerosol types and different aerosol cases (e.g. dust or non-dust cases)

**Page 15.8:** Comma added.

**Page 26.1:** "Under" is deleted.

**Page 26.2:** Capital "T" is used according to the reviewer's correction.

**Page 26.3:** Clarified. "Climatological" is added.

**Page 26.4:** Indeed the two figures it is better to be plotted together, the authors totally agree with the reviewer's recommendation.

**Page 26.5:** Corrected. Word "Geographical" is deleted from the caption of figures 5, 6 and 7.

**Page 27.1:** Both the scale and the colormap are modified, according to the suggestion by the reviewer. Please see the figures below, before and after the adaptation of the figures.

**Page 27.2:** Both the scale and the colormap are modified, according to the suggestion by the reviewer. Please see the figures below, before and after the adaptation of the figures. In addition the phrase "please note the different height scale" is added to the manuscript.

**Before**

[Figure]

Fig. 4: Spatial distribution of dust occurrence [%], climatological pure-dust CoM (Center of Mass) and dust TH (Top Height) in km a.g.l., for each season over the domain between 65°-155° E and 5°-55° N and for the period 01/2007-12/2015.

**After**

[Figure]

**Fig. 4: Spatial distribution of dust occurrence [%], climatological pure-dust CoM (Center of Mass) and dust TH (Top Height) in km a.g.l., for each season over the domain between 65°-155° E and 5°-55° N and for the period 01/2007-12/2015.**

**Page 30.1:** Since CALIPSO CALIOP classifies layers as dust or polluted dust over these domains, even if the frequency is low - as indicated by the question of the reviewer, it makes sense to decouple those few cases to the pure-dust and non-dust components of the detected aerosol mixtures. The need is supported also by the evidence that dust sources are not only natural but anthropogenic activity may also lead to dust particles in the atmosphere, over domains where no natural dust is supposed to be observed through intercontinental transport. Chen, S., Huang, J., Jiang, N., Zang, Z., Guan, X., Ma, X., Jia, Z., Zhang, X., Zhang, Y., Huang, K., Xu, X., Zhang, G., Li, J., Yang, R., and Liao, S.: Estimations of anthropogenic dust emissions at global scale from 2007 to 2010, Atmos. Chem. Phys. Discuss., https://doi.org/10.5194/acp-2017-890, in review, 2017.

**Page 30.2:** Suggestion is implemented – comment 27.1.

**Page 30.3:** The scale of figures are adapted according to the recommendation of the reviewer in order to include both higher values (comments 27.1, 27.2, 30.3)

**Page 31.1:** The authors are of the opinion that any redundant information added on the figures 5, 6 and 7 would make it harder to read features on them. Thus the authors have not included marks indicating the desert regions on the figures, since a reader can take the information of the regions of interest and the locations of the deserts from Figure 1.

**Page 32.2:** According to the reviewer's recommendation, the caption of the colormap is changed to: "Dust Extinction Coefficient".

[revised manuscript text omitted]

5   **Fig.2**

**Page: 46**

Number: 1    Author: Referee    Subject: Sticky Note    Date: 20.12.2017 12:30:24
Letters are only half seen here...

[Figure]

**Fig.3**

[Figure]

Fig. 4

[Figure]

**Fig. 5**

[Figure]

Dec-Jan-Feb | Mar-Apr-May | Jun-Jul-Aug | Sep-Oct-Nov

Height (km)

Longitude (deg)

NaN 0   1   2   5   10   20   25   30   35   40   45   50   75   100   125   150   175   200

Dust Extinction Coefficient 532 nm (km$^{-1}$)

5    **Fig. 6**

[Figure]

Dec-Jan-Feb  Mar-Apr-May  Jun-Jul-Aug  Sep-Oct-Nov

Particle Depolarization Ratio

5    **Fig. 7**

[Figure]

**Fig. 8**

[Figure]

**Fig. 9**

---

## Author Response (AR2)

**Response to Anonymous Referee #2**

The authors would like to thank once more the referee for his time, for revising the submitted manuscript and for the suggestions. We have incorporated the proposed changes and corrections in the revised manuscript. Following, you will find our responses one-by-one that are addressed to the Editorial board and the reviewers as well.

**Comments**

**Page 7.1:** According to the reviewer's suggestion the following sentence is added to the Data and Methodology: "The discrimination of the pure-dust component from the total aerosol load is a polarization-based technique, therefore it is possible to provide the global horizontal and vertical distribution of pure-dust only through satellite-based active remote sensing (CALIPSO CALIOP, ISS-CATS (McGill et al., 2015)). "

**Page 7.2:** According to the suggestion the text is modified: from "… implemented …" to "… used …"

**Page 7.2:** Suggestion is included.

**Page 7.2:** The sentence is modified, though the authors would this information on not to be removed from the conclusion section.

**Page 8.1:** Suggestion is included.
**Page 8.2:** Suggestion is included.
**Page 7.2:** Suggestion is included.
**Page 7.2:** Suggestion is included.

**Page 9.1: e 2.2:** The text is corrected according to the reviewer.

**Page 10.1:** Suggestion is included.

**Page 11.1:** Suggestion is included.

**Page 12.1:** Suggestion is included.

**Page 23.1:** Suggestion is included.

**Page 24.1:** Figure is modified.

[revised manuscript text omitted]

CALIPSO L2 Aerosol Products
**Main Parameters**
Backscatter and Depolarization at 532 nm
**Auxiliary Parameter Parameters**
AVD, QC Flags, CAD Score, Temperature, Uncertainties, Surface Elevation, Geolocation, Time

Quality Filtering of L2 Profiles
(Winker et al., 2013; Marinou et al., 2017)

L2 filtered CALIPSO Profiles

Dust aerosols?

Yes

No

$$\beta_1 = \beta_t \frac{(\delta_p - \delta_2)(1 + \delta_1)}{(\delta_1 - \delta_2)(1 + \delta_p)}$$

(Tesche et al., 2009)

Non-Dust Component

Accept as is

Dust Component of aerosol mixture

Apply appropriate dust LR

CALIPSO-based Pure-Dust product (Amiridis et al., 2013)

Apply appropriate LR per aerosol type (Omar et al., 2009)

Non-Dust Component of aerosol mixture (Marine, Clean Continental, Polluted Continental, Smoke)

Pure-Dust Extinction Coefficient Profiles

Non-Dust Extinction Coefficient Profiles

Value of NaN assigned to the non-dust aerosol types when averaging over a grid.

Value of 0 km$^{-1}$ assigned to the non-dust aerosol types when averaging over a grid.

Conditional Dust Extinction Coefficient Product

Climatological Dust Extinction Coefficient Product

Total Extinction Coefficient

5    **Fig.2**

[Figure]

**Fig.3**

[Figure]

**Fig. 4**

[Figure]

**Fig. 5**

[Figure]

**Fig. 6**

[Figure]

**Dec-Jan-Feb**  **Mar-Apr-May**  **Jun-Jul-Aug**  **Sep-Oct-Nov**

Particle Depolarization Ratio

**Fig. 7**

[Figure]

**Fig. 8**

[Figure]

Fig. 9